METHODS AND RESOURCES

# Single-cell in vivo imaging of cellular circadian oscillators in zebrafish

Haifang Wang[1,2☯], Zeyong Yang[3,4,5☯], Xingxing Li[6☯], Dengfeng Huang[1,2], Shuguang Yu[1,2], Jie He[1,2*], Yuanhai Li[6*], Jun Yan[1,2,7*]

**1** Institute of Neuroscience, State Key Laboratory of Neuroscience, CAS Center for Excellence in Brain Science and Intelligence Technology, Chinese Academy of Sciences, Shanghai, China, **2** Shanghai Center for Brain Science and Brain-Inspired Intelligence Technology, Shanghai, China, **3** Department of Anesthesiology, International Peace Maternity and Child Health Hospital, Shanghai Jiao Tong University School of Medicine, Shanghai, China, **4** Shanghai Key Laboratory of Embryo Original Disease, Shanghai, China, **5** Shanghai Municipal Key Clinical Specialty, Shanghai, China, **6** Department of Anesthesiology, First Affiliated Hospital of AnHui Medical University, Hefei, Anhui, China, **7** School of Future Technology, University of Chinese Academy of Sciences, Beijing, China

☯ These authors contributed equally to this work.

\* junyan@ion.ac.cn (JY); liyuanhai-1@163.com (YL); jiehe@ion.ac.cn (JH)

**Data Availability Statement:** All the imaging data were included in the supplementary data files S1, S2 Data and S1–S32 Movies; the single-cell RNA-seq data have been deposited on GEO (GSE134288).

## Abstract

The circadian clock is a cell-autonomous time-keeping mechanism established gradually during embryonic development. Here, we generated a transgenic zebrafish line carrying a destabilized fluorescent protein driven by the promoter of a core clock gene, *nr1d1*, to report in vivo circadian rhythm at the single-cell level. By time-lapse imaging of this fish line and 3D reconstruction, we observed the sequential initiation of the reporter expression starting at photoreceptors in the pineal gland, then spreading to the cells in other brain regions at the single-cell level. Even within the pineal gland, we found heterogeneous onset of *nr1d1* expression, in which each cell undergoes circadian oscillation superimposed over a cell type–specific developmental trajectory. Furthermore, we found that single-cell expression of *nr1d1* showed synchronous circadian oscillation under a light–dark (LD) cycle. Remarkably, single-cell oscillations were dramatically dampened rather than desynchronized in animals raised under constant darkness, while the developmental trend still persists. It suggests that light exposure in early zebrafish embryos has significant effect on cellular circadian oscillations.

## Introduction

Circadian rhythm evolves to align animal behaviors to periodic daily environmental changes. At the molecular level, the vertebrate circadian clock is mainly generated through transcriptional/translational feedback loops of core clock genes [1]. Among them, two transcription factors (TFs), BMAL1 (also known as ARNTL or MOP3) and CLOCK form heterodimers to bind to E-boxes in the promoters and initiate the transcription of their target genes [2–4], including Per family genes (*Per1*, *Per2*, and *Per3*) and Cry family genes (*Cry1* and *Cry2*). The activation of these genes results in the formation of the PER/CRY complex and thereby inhibits CLOCK/

**Funding:** This work was supported by National Science Foundation for Young Scientists of China grant (No. 31701029) and Natural Science Foundation of Shanghai grant (16ZR1448800) to HW; NSFC-ISF Joint Scientific Research Program grants (31861143035) to JY; and Natural Science Foundation of China grants (No. 31571209 to JY and No. 81401279 to ZY). The funders had no role in study design, data collection and analysis, decision to publish, or preparation of the manuscript.

**Competing interests:** The authors have declared that no competing interests exist.

**Abbreviations:** CCD, charge-coupled device; CT, circadian time; DD, constant dark; dpf, days postfertilization; DsRed, red fluorescent protein from *Discosoma*; E, embryonic day; ESC, embryonic stem cell; FUCCI, fluorescent, ubiquitination-based cell cycle indicator; GCaMP, genetically encoded calcium indicator; iPSC, induced pluripotent stem cell; LD, light–dark; LoG, Laplacian of Gaussian; NA, numerical aperture; P, postnatal day; PA, poly(A) site; SCN, suprachiasmatic nucleus; scRNA-seq, single-cell RNA-seq; SNN, shared nearest-neighbor; t-SNE, *t*-distributed stochastic neighbor embedding; TF, transcription factor; TGFPD1, peTurboGFP-dest1; Tol, transposable element of *Oryzias latipes*; VNP, Venus-NLS-PEST; ZT, Zeitgeber time.

BMAL1 transcriptional activity, forming a negative feedback loop [5]. Nuclear receptor, REV-ERBα (also known as NR1D1), represses the transcription of *Bmal1* and itself is under the transcriptional regulation of BMAL1/CLOCK, giving rise to the second negative feedback loop of the circadian clock [6]. The genome-wide regulation by circadian TFs such as BMAL1/CLOCK and REV-ERBα typically leads to thousands of genes showing circadian expression in a given tissue. Although the basic network of core circadian genes is present in almost every cell, many of the circadian-controlled genes are tissue specific or cell type specific. Their circadian expression is a result of either tissue-specific binding of circadian TFs [7] or transcriptional cascade from tissue-specific TFs regulated by circadian TFs [8]. At the organismal level, the overt circadian rhythm is governed by an intricate network of circadian oscillators in which the master pacemaker, such as suprachiasmatic nuclei (SCN) in mammals or the pineal gland in zebrafish, is believed to play a pivotal role [1,9]. Traditional in vivo or ex vivo studies of the cellular circadian clock have relied on luciferase reporter systems driven by core clock gene promoters [10,11]. But luciferase imaging can only achieve single-cell resolution in organotypic slices in culture with a high-resolution charge-coupled device (CCD) camera. Transgenic zebrafish lines carrying luciferase reporters driven by core clock gene promoters have been developed in larval zebrafish. But in vivo luciferase imaging of these fish lines lacked single-cell resolution and can only report the population-level circadian rhythm [10,12]. Genetically encoded calcium indicator (GCaMP) has been widely used to monitor in vivo single-cell calcium activity. In comparison, there is still a lack of zebrafish line to report the circadian expression at the single-cell level in vivo.

Circadian rhythm has to be established at molecular, cellular, tissue, and behavioral levels during animal development. Day-night rhythms in the fetal rat SCN are first detected between embryonic day E19 and E21 [13]. Rhythmic expression of circadian clock genes is not detected in both in vitro and in vivo mouse embryonic stem cells (ESCs) but only appears when ESCs differentiate into neural stem cells [14,15]. By ex vivo luciferase imaging of mouse fetal SCN, Carmona-Alcocer and colleagues showed that a few cells in SCN start circadian oscillations on E14.5, widespread synchronized oscillations were formed on E15.5, and then a dorsal-ventral phase wave was established at postnatal day P2 [16]. Zebrafish (*Danio rerio*) embryos that are in vitro fertilized and transparent provide an accessible model organism for in vivo live imaging, and thereby are being widely used in the study of animal development. A functional circadian clock, characterized by free-running activity, rhythmic cell cycle, and circadian gene expression, is established after hatching in zebrafish [17]. However, how single-cell circadian clocks were established during embryonic development is still unclear. It is well documented that early exposure to a light–dark (LD) cycle is required for the development of the circadian clock in zebrafish larva [10]. But it is still under debate whether the effect of light on clock development is a synchronization of already existing oscillators or an initiation of single-cell clocks. Questions like these can only be addressed by time-lapse in vivo imaging of single-cell circadian clock reporter.

Here, we report a transgenic zebrafish line using destabilized fluorescent protein, Venus-NLS-PEST (VNP), driven by the promoter of a key circadian clock gene, *nr1d1*. This system allows us to monitor the development of single-cell circadian rhythm in live zebrafish larva in a cell type–specific manner. We observed that VNP reporter expression undergoes stepwise onset starting at the photoreceptor cells in the pineal gland, then spreads to cells in other brain regions. Using single-cell RNA-seq (scRNA-seq), we characterized the cell types expressing VNP in the whole brain. Within the pineal gland, we found that each cell undergoes circadian oscillation superimposed over cell type–specific developmental trajectories. Under LD cycle, cellular expression of VNP shows synchronous circadian oscillation. However, the circadian expression of *nr1d1*:VNP-positive cells was dramatically dampened rather than

desynchronized, while the developmental trend was still present at the single-cell level in fish raised under constant darkness. Our result suggests that the early exposure of LD cycle is crucial for the ontogeny of functional single-cell oscillators.

## Results

### Screening for in vivo circadian reporters in zebrafish

To monitor circadian rhythm at the single-cell level in live animals, we have screened for in vivo circadian reporters among various combinations of destabilized fluorescent proteins driven by core clock gene promoters in larval zebrafish. We first tested peTurboGFP-dest1 (TGFPD1) encoding a destabilized variant of green fluorescent protein TurboGFP. The plasmids of *bmal1a/bmal2/per2/nr1d1*:TGFPD1 were constructed by homologous recombination. We observed that none of these plasmids were expressed in F1 embryos. We then tested the plasmids containing core clock gene promoters driven VNP, another form of destabilized fluorescent protein. We found that the plasmids of *cry2b/bmal1a/per2*:VNP were not expressed in F1 zebrafish embryos, while *per1a*:*VNP* has too low expression to be used for in vivo imaging. For *nr1d1*:VNP, it has been reported that zebrafish *nr1d1* gene has two promoters, i.e., ZfP1 (distal) and ZfP2 (proximal). ZfP1 is conserved and functionally similar to mammalian *Nr1d1* promoter [18]. We found that the F1 fish containing only the proximal *nr1d1* promoter ZfP2 (1.5 kbp) failed to drive VNP expression. But *nr1d1*:VNP containing both ZfP1 and ZfP2 (6.2 kbp) showed robust expression in zebrafish embryos (Fig 1A and Table A in S2 Data). Therefore, we chose *nr1d1*:VNP containing ZfP1 and ZfP2 promoters as the circadian reporter and generated the transgenic fish line Tg(*nr1d1*:VNP) by the transposable element of *Oryzias latipes* (Tol) 2 system.

The 6.2-kbp ZfP1 and ZfP2 promoter region to drive VNP expression includes the entire set of known *cis*-regulatory elements: E-box, RRE, Crx-, and Otx5-binding sites (Fig 1A). We measured the mRNA levels of endogenous *nr1d1* and VNP expression in the Tg(*nr1d1*:VNP) transgenic fish from 3.5 days postfertilization (dpf) to 7.5 dpf under the 12-hour/12-hour LD cycles using real-time PCR. *nr1d1* and VNP showed highly correlated expression patterns (Pearson's r = 0.9) indicating that VNP expression faithfully reported the expression of *nr1d1* at the mRNA level. In addition, both genes showed a higher expression level at the dawn than dusk over days (Fig 1B), which is consistent with our previous result that *nr1d1* mRNA shows circadian expression peaking at ZT23 (Zeitgeber time) in larval zebrafish [8]. Previous studies have shown that the rhythmic *period* gene expression starts from the second day after birth [19,20]. We found that *nr1d1* mRNA also starts oscillating from the second day of the development, as early as the *per1b* gene (S1A Fig). Therefore, *nr1d1* reporter exhibits similar developmental profiles as *per1b* and can be used for studying when and how the cellular clock is developed.

We examined the spatial distribution of fluorescence-labeled cells in *nr1d1*:VNP at 7.5 dpf using an in vivo two-photon imaging system and found *nr1d1*:VNP-positive cells in many brain regions, including the pineal gland, the optic tectum, and the cerebellum (Fig 1C, S1 and S2 Movies). In particular, we observed that *nr1d1*:VNP-positive cells are most highly expressed in the pineal gland. We next monitored in vivo expression of *nr1d1*:VNP at the whole-brain scale from 3.5 dpf to 7.5 dpf at ZT0 and ZT12 (Fig 1D) by time-lapse imaging. We aligned the time series of 3D images using CMTK toolkit (see Methods) so that we can trace each single cell in the same fish over time. In this way, we observed that the expression of *nr1d1*:VNP was sequentially turned on and underwent gradual increase in distinct brain regions during development, as early as 3.5 dpf in the pineal gland, followed by the optic tectum at 5.5 dpf, and other brain regions such as the cerebellum at later time points (Fig 1E, Table B in S2 Data, S1

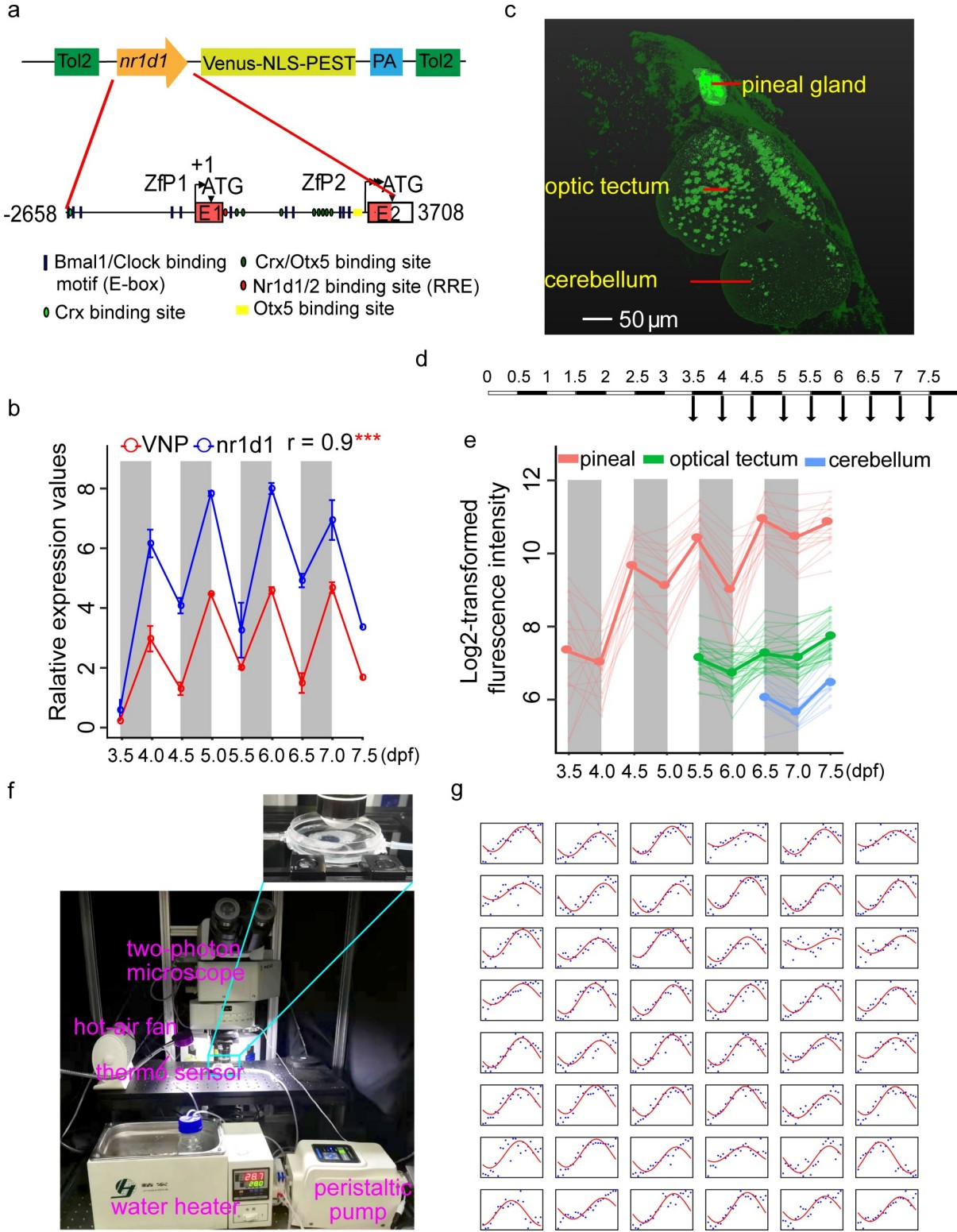

**Fig 1. Construction of in vivo circadian reporter in zebrafish.** (a) The upper graph shows the schematic of *nr1d1*:VNP construct design. The lower graph shows a magnified view of the *nr1d1* promoter sequence used for driving the circadian expression of VNP. The putative RRE (Nr1d1/2 binding site), E-box (Bmal1/Clock binding site), Crx, Otx5, and Crx/Otx5 binding sites were indicated by red oval, blue rectangle, light green oval, yellow rectangle, and dark green oval, respectively. (b) Plot of real-time PCR results of *nr1d1* and *Venus-pest* expression. Each time point has 2 replicates, and each replicate is the pool of 7–10 fish. The dots show the original value, while the solid lines and the bars

represent mean ± SEM. (c) Whole-brain two-photon image shows the spatial distribution of *nr1d1*:VNP-positive cells in different brain regions at 7.5 dpf. (d) Experimental design to examine the developmental dynamics of *nr1d1*:VNP expression in the whole zebrafish brain. (e) Single-cell tracing result of *nr1d1*:VNP-positive cells in the pineal gland (red), the optic tectum (green), and the cerebellum (blue) for one fish. The cells were tracked from 3.5 dpf in the pineal gland, 5.5 dpf in optic tectum, and 6.5 dpf in cerebellum, respectively. Each thin line represents one cell, while the thick line represents the mean value of cells in each brain region. (f) Illustration of the setup for time-lapse imaging. The fish were fixed in a chamber with circulating egg-water without anesthesia at temperature of 28 ± 0.5˚C. (g) Single-cell tracing results of all 48 *nr1d1*:VNP-positive cells imaged at 1-hour resolution. The blue dots represent the original fluorescence signals, while the solid red line represents the smoothed curve fitted by the cosine functions. The numerical values for panels b, e, and g are in S1 Data. dpf, days postfertilization; PA, poly(A) site; Tol2, transposable element of *Oryzias latipes* 2; VNP, Venus-NLS-PEST.

and S2 Movies). In addition, single-cell tracing revealed that circadian oscillations of *nr1d1*:VNP expression appear to peak at ZT12 across brain regions (Fig 1E). The expression pattern of *nr1d1*:VNP closely resembled that of endogenous *nr1d1* reported by Delaunay and colleagues [19]. Thus, *nr1d1*:VNP fish can be used as a reporter for both developmental expression and circadian expression of endogenous *nr1d1* gene in zebrafish. To perform high temporal resolution imaging, we imaged the fish every hour for one day starting at 5 dpf (Fig 1F and 1G, S3 Movie, Table C in S2 Data) by a time-lapse imaging system, as illustrated in Fig 1F. The fishes were embedded in agarose gel and kept alive during the entire imaging process. We observed that 39 out of 48 traced cells in the pineal gland showed circadian oscillations using JTKcycle adjusted $P < 0.05$ and absolute oscillating amplitude >100 as the cutoff (Fig 1G and S1B and S1C Fig). This shows that *nr1d1*:VNP fish can report cellular circadian oscillations under continuous imaging.

## Characterization of *nr1d1*:VNP-expressing cells

To identify the cell types expressing *nr1d1*:VNP in the whole brain, we conducted single-cell RNA-seq (scRNA-seq) of approximately 15,000 cells dissociated from the brain of Tg(*nr1d1*: VNP) larval fish at 6.5 dpf. Among them, 6,514 cells detected with more than 500 genes were used for the downstream analysis. In total, 26 cells clusters were classified from scRNA-seq. They were manually annotated by comparing the marker genes with those from whole-brain scRNA-seq data in adult zebrafish [21] (Fig 2A). Wilcoxon Rank Sum test implemented in Seurat package was applied to calculate the significance of enrichment of VNP signal in each brain region. Using adjusted $P < 0.01$ as the cutoff, we found that the mRNA of *nr1d1*:VNP was most highly enriched in cell clusters of photoreceptors in the pineal gland, which is consistent with the imaging data and leads us to focus on the pineal gland in the following analysis. The mRNA of *nr1d1*:VNP was also enriched in glutamatergic neurons in forebrain and hypothalamus, granule cells in cerebellum, dorsal habenula cells, as well as some non-neuron cells such as oligodendrocyte, retinal-pigment epithelium-like cells, and endothelial cells (Fig 2B).

We next examined the expression of *nr1d1*:VNP in the pineal gland at the single-cell resolution. Fig 2C showed an example of 3D reconstruction of *nr1d1*:VNP signals in the pineal gland. At 4.5 dpf, each pineal gland contains about 60 *nr1d1*:VNP-positive cells. To better understand the cell type of these cells, we crossed Tg(*nr1d1*:VNP) with Tg(*aanat2*:mRFP) [12]. Averaged 3D cell density distribution of *nr1d1*:VNP-positive cells across six fish (Methods section) revealed that *nr1d1*:VNP-positive cells were mostly distributed around the lumen of the pineal gland, while the cell density was relatively low in the central part of the pineal gland (Fig 2D). We observed that *nr1d1*:VNP signals were highly overlapped with the *aanat2*:mRFP signals (94%) indicating that most *nr1d1*:VNP-positive cells were melatonin-synthesizing photoreceptor cells in the pineal gland (Fig 2C, S4 Movie). According to our scRNA-seq data, VNP is expressed in both rod-like photoreceptors and cone-like photoreceptors (Fig 2E). Indeed, we observed co-labeling of VNP with rod cell marker (*xops*) (10%) and cone cell marker (*lws2*)

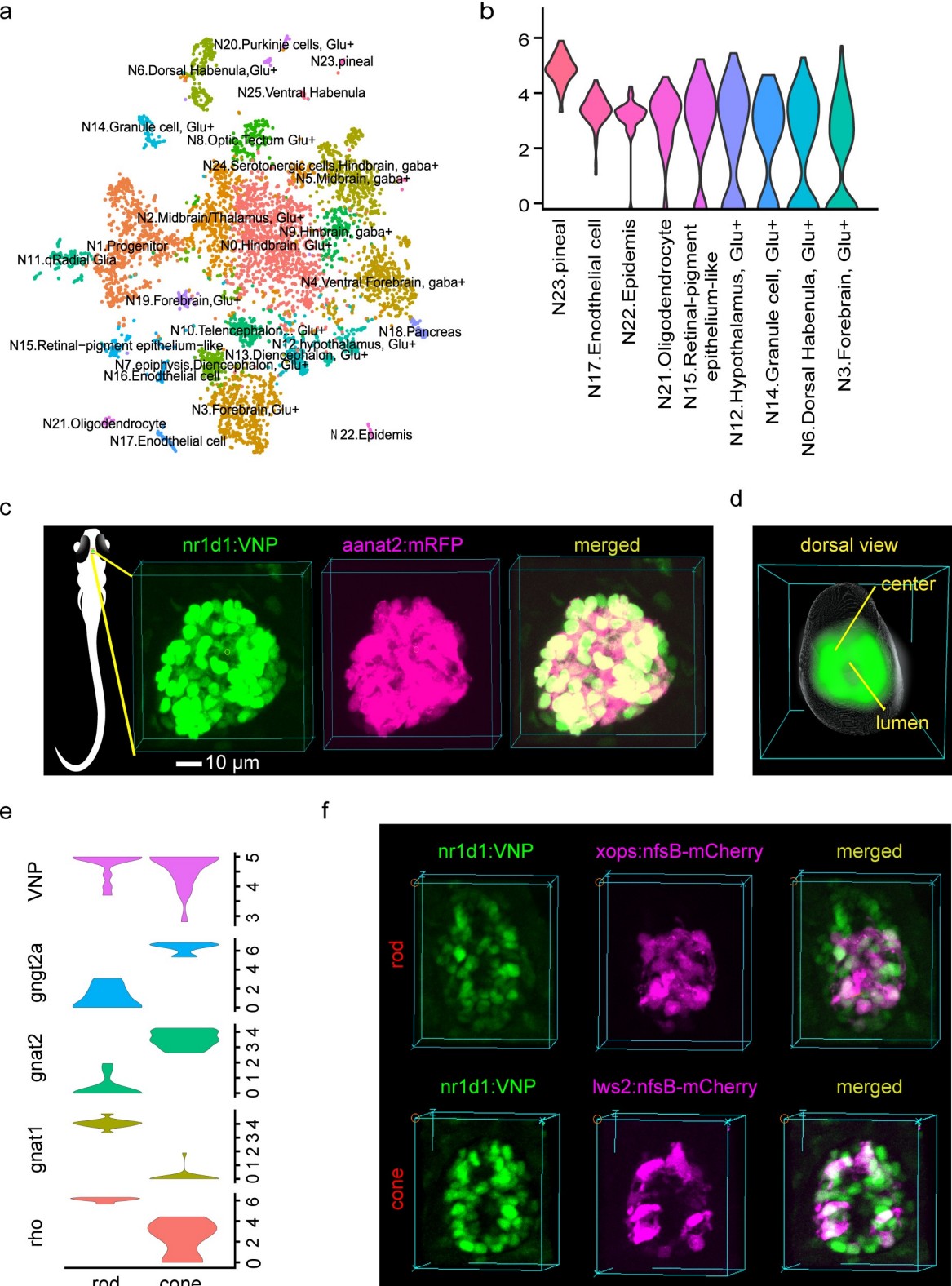

**Fig 2. Characterization of *nr1d1*:VNP expressing cells.** (a) t-SNE visualization of brain cell clusters. The clusters were annotated by comparing to the adult zebrafish scRNA-seq data. (b) Violin plot demonstrated the clusters with enriched expression of *nr1d1*:VNP. *Y* axis denotes the normalized expression value. Note the highest levels in the pineal gland. (c) Fluorescence images show the co-expression of

*nr1d1*:VNP (nuclear signal) and *aanat2*:mRFP (cytoplasmic signal) in the zebrafish pineal gland. The left graph illustrates the 3D location of the pineal gland in zebrafish larva. (d) Three-dimensional reconstruction of a common zebrafish pineal gland by aligning and averaging of six fish. The gray sphere represents the boundary of the pineal gland and the green color represents the density distribution of *nr1d1*: VNP-positive cells in the pineal gland. (e) Violin plot showed the expression of rod cell markers (*rho*, *gnat1*), cone markers (*gnat2*, *gngt2a*), and *nr1d1*:VNP in photoreceptor clusters. *Y* axis denotes the normalized expression value. (e) Fluorescence images showed the co-expression of *nr1d1*:VNP (nuclear signal) with the Tg(*xops*:nfsB-mCherry) fish line (cytoplasmic signal) and Tg(*lws2*:nfsB-mCherry) fish line (cytoplasmic signal) in the zebrafish pineal gland. The numerical values for panel b and e are in S1 Data. scRNA-seq, single-cell RNA-seq; t-SNE, *t*-distributed stochastic neighbor embedding; VNP, Venus-NLS-PEST.

(21%) in the pineal gland by crossing the Tg(*nr1d1*:VNP) fish line with Tg(*xops*:nfsB-mCherry) and Tg(*lws2*:nfsB-mCherry) fish lines, respectively (Fig 2F). Our scRNA-seq data also suggested that *nr1d1*:VNP is expressed in proliferative cells. Indeed, when we crossed our VNP fish line with Tg(*her4*:DsRed) zebrafish marking proliferative cells [22], we found co-labeling of VNP and red fluorescent protein from *Discosoma* (DsRed) in the pineal at 5.5 dpf (26%) (S1D Fig). This result indicated that a small number of photoreceptors could be still pro-liferating at 5 dpf. The co-labeled cells may include a small number of glial cells, which also express high levels of *her4*, as scRNA-seq data suggested that *nr1d1* is also expressed in non-photoreceptor cells such as glial cells.

## Developmental dynamics of *nr1d1*:VNP expression within the pineal gland

To examine the development of circadian oscillations more closely in the pineal gland, we imaged *nr1d1*:VNP signals in the pineal gland from 3.5 dpf to 6.5 dpf using two-photon imaging (Fig 3A and 3B, S5–S10 Movies, Table D in S2 Data). The time series of 3D images were first aligned using CMTK toolkit. Then, the fluorescent intensities of each single cell in the aligned images were traced by Trackmate (a Fiji plug-in) [23] (details in Methods). The majority of cells already showed higher *nr1d1*:VNP signals in dusk than dawn when the signals first become detectable in the pineal gland (S2 Fig). This pattern does not change significantly during devel-opment (ANOVA *P* = 0.78) (S2 Fig). However, we found that VNP-positive cells within the pineal gland display heterogeneous temporal patterns of *nr1d1*:VNP expression over develop-ment (Fig 3C). Some cells showed rapid increase in the baseline level, while other cells showed more robust circadian oscillations. We then quantified the circadian and developmental com-ponents in each cell by fitting the time-series data with a regression model that combines a step-wise function of time, $(B*(-1)^{(x+1)})$, denoting circadian oscillation, with a linear function of time, $(Ax + C)$, denoting the developmental increase (Fig 3D). After fitting the model, we found that the regression coefficients of developmental effect (A) and circadian oscillation (B) both vary widely across cells, but there is a negative correlation between them (Fig 3E).

To reveal if the heterogeneity of single-cell temporal profiles within the pineal gland is due to the differences in cell type, we next conducted cell type–specific imaging by crossing *nr1d1*:VNP with the fish line, labeling rod-like cells in red fluorescent protein Tg(*xops*:nfsB-mCherry). After imaging *nr1d1*:VNP signals in the rod-like and non-rod-like cells simulta-neously in the same fish during development from 3.5 dpf to 6.5 dpf (Table E in S2 Data, S11–S16 Movies), we found that rod-like cells have higher level of baseline expression than non-rod-like cells (Fig 3F and 3G). Taken together, pineal photoreceptor cells of zebrafish lar-vae undergo circadian oscillations superimposed over cell type–specific developmental trajectories.

## Single-cell circadian oscillations in the pineal gland

We next monitored the *nr1d1*:VNP signals in the pineal glands of live zebrafish larvae with temporal resolution every 2 hours at 5 dpf using two-photon imaging (Fig 4A, S17 and S18

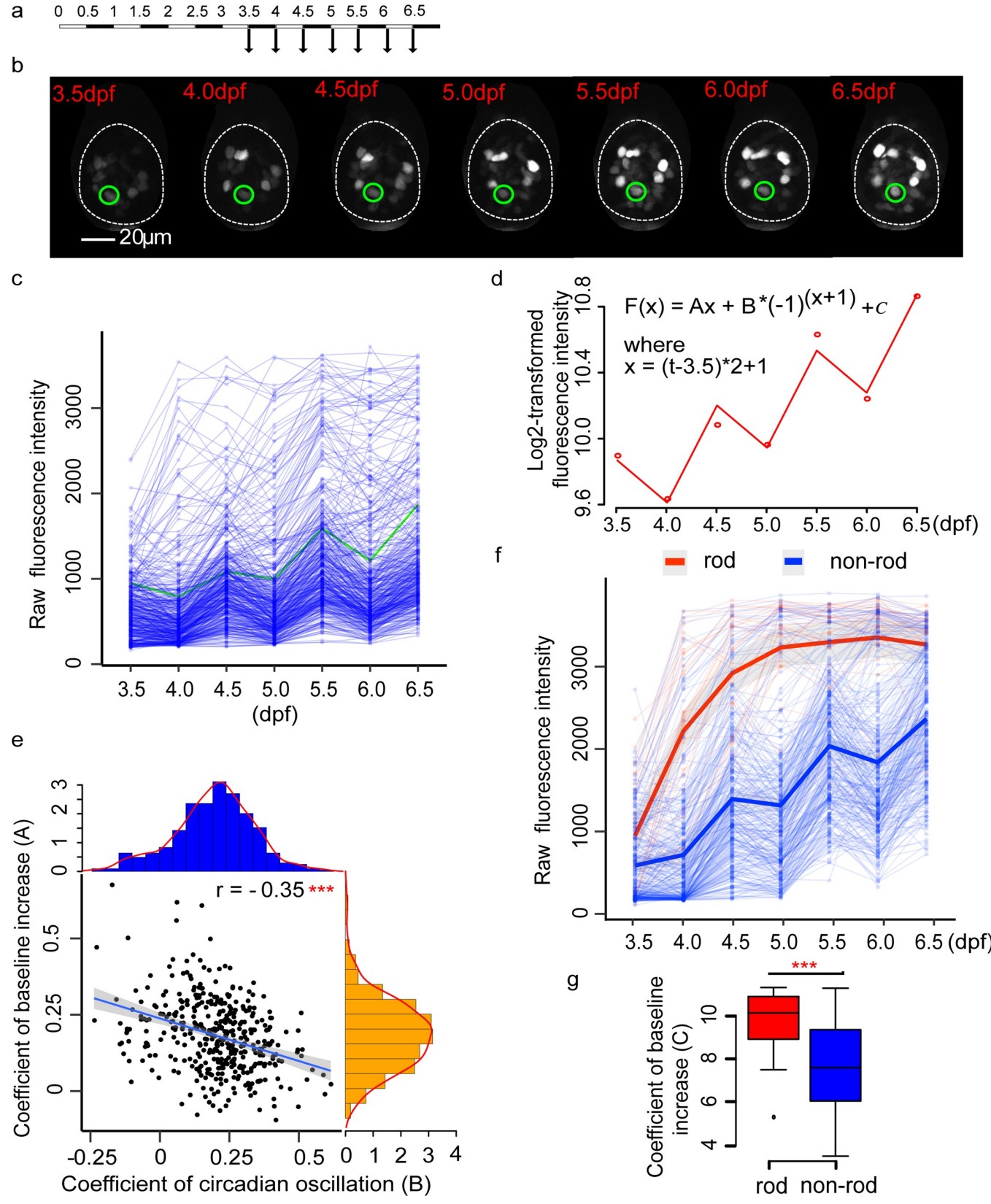

$$F(x) = Ax + B*(-1)^{(x+1)} + C$$

where
$$x = (t-3.5)*2+1$$

**Fig 3. Developmental dynamics of *nr1d1*:VNP expression.** (a) Experimental design to examine the developmental dynamics of *nr1d1*:VNP expression. (b) Fluorescence images illustrate the tracing of one *nr1d1*:VNP cell during development from 3.5 dpf to 6.5 dpf. This cell was highlighted in green in Fig 3C. (c) Raw fluorescence intensity of the traced cells (six fish). (d) Regression model used to fit the fluorescence intensity (the cell is illustrated in Fig 3B and highlighted in Fig 3C). The dots represent the fluorescence intensity, while the line represents the fitting curve. (e) Scatterplot demonstrated the relationship between coefficients of developmental effect (A) and circadian oscillation (B). The histograms in orange and blue showed the distribution of coefficients of developmental effect (A) and the distribution of coefficients of circadian oscillations (B), respectively. ***Pearson's correlation $P < 0.001$. (f) Raw fluorescence intensity of rod-like cells and non-rod-like cells (six fish). Each thin line represents one cell and each dot represents raw fluorescence intensity. The thick lines represent the loess-smoothed curves for all rod-like cells in red and non-rod-like cells in blue, respectively. The shaded areas show the 95% confidence level of the smoothed curve. (g) A comparison of baseline expression (C) between rod-like and non-rod-like cells. Two-tailed Student *t* test was applied to calculate the levels of significance between the two types of cells. **$P < 0.01$. The numerical values for panels c–g were in S1 Data. dpf, days postfertilization.

Movies). We traced every *nr1d1*:VNP-positive cell at a different time of the day (Fig 4B, Table F in S2 Data). Fig 4C showed the trace plots of individual cells of two zebrafish throughout the day. JTKcycle was applied to calculate the circadian phase and amplitude of each cell. Using JTKcycle adjusted $P < 0.05$ and absolute oscillating amplitude >100 as the cutoff, we identified 99 cells showed circadian oscillations in a total of 117 *nr1d1*:VNP-positive cells in the pineal gland. Their circadian phases were distributed around ZT12 within a narrow range (Fig 4D), with Kuramoto order parameter R = 0.86 indicating the phase coherence of single-cell circadian oscillators [24]. Clustering analysis of single-cell VNP traces identified two distinct clusters of cells that can be distinguished by their baseline fluorescence intensities (Fig 4E and 4G). Such difference in baseline level of expression at 5 dpf can be explained by the cell type–specific developmental trajectories that we observed above. We also observed that the cluster with higher baseline level (cluster 2) also showed lower relative amplitude of oscillation (Fig 4H). Similar negative correlation between mean transcriptional activity and relative amplitude of circadian oscillations has been reported in mouse [25]. However, there is no significant difference in circadian phase between the two clusters (S3 Fig). In short, circadian clocks in pineal photoreceptor cells are oscillating synchronously under LD in spite of the large difference in baseline level.

## LD cycle is essential for cellular clocks

It is known that LD cycle is required for the development of the circadian clock in zebrafish [10]. However, it is unclear whether the role of LD cycle in clock ontogeny is a synchronization of individual oscillators or an initiation of oscillation. To address this question, we imaged *nr1d1*:VNP signals in zebrafish larvae raised under constant dark (DD) from 3.5 dpf to 6.5 dpf using two-photon imaging (Fig 5A, S19–S22 Movies, Table G in S2 Data). Examination of the expression patterns of all *nr1d1*:VNP-positive cells from four DD fish showed that, although the expression of *nr1d1*:VNP still increased during development, no obvious oscillations were observed in DD cells (Fig 5B). Indeed, regression analysis showed that the oscillation coefficients defined as the stepwise function of the majority of DD cells are not significantly different from zero (Fig 5C). However, DD cells have even higher developmental coefficients than LD cells (Fig 5D). In comparison, when we imaged the fish transferred into DD condition after six LD cycles (LD_DD cells) (Fig 5E–5G, Tables H–I in S2 Data, S23–S28 Movies), we observed that LD_DD cells were still oscillating into two days of darkness (Fig 5F) and the amplitude of circadian oscillation showed no significant difference compared to the fish kept under LD condition (LD_LD cells) (Fig 5G). To further examine the circadian oscillation in dark-raised fish, we imaged *nr1d1*:VNP signals in the pineal gland every 2 hours at 5 dpf (Fig 5H, Table J in S2 Data, S29–S31 Movies). JTKcycle (from MetaCycle package in R) was applied to calculate the circadian phase and amplitude of each cell. We found that DD cells showed much lower oscillating amplitudes than those in LD cells (Fig 4I and 4J). Using JTKcycle adjusted $P < 0.05$ and absolute amplitude >100 as the cutoff, only 3 out of 142 DD cells showed significant

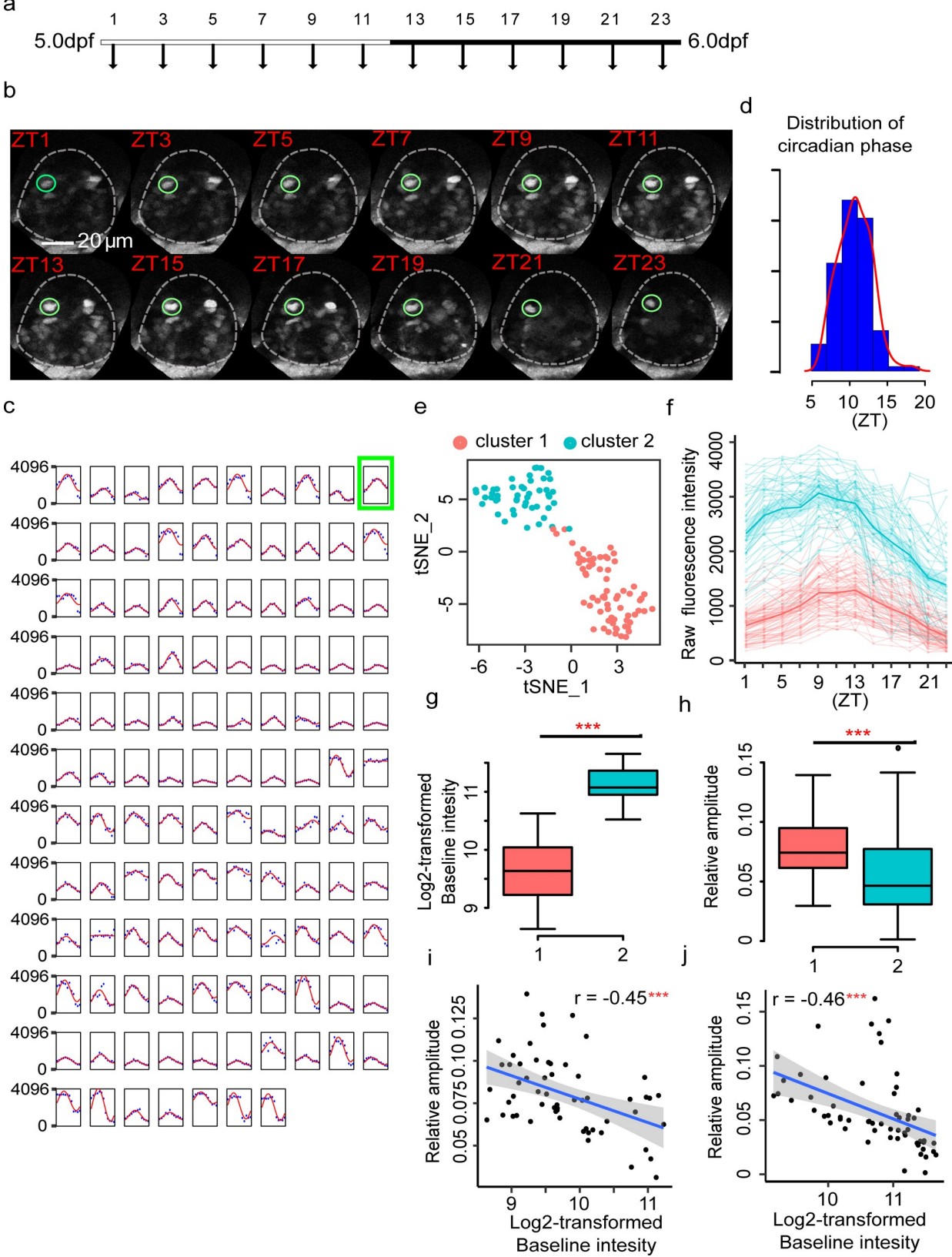

**Fig 4. Circadian dynamics of the pineal gland at higher temporal resolution (two fish).** (a) Experimental design to examine the developmental dynamics of *nr1d1*:VNP expression at higher temporal resolution. (b) Fluorescence images illustrate the tracing of one *nr1d1*:VNP example cell every 2 hours. (c) Single-cell tracing results of all 117 *nr1d1*:VNP-positive cells in two zebrafish pineal glands (two fish). The blue dots represent the original fluorescence signals, while the solid red line represents the smoothed curve fitted by the cosine functions. The example cell in (b) is highlighted by a green square. (d) Circadian phase distributions of the 117 *nr1d1*:VNP-positive cells. (e) t-SNE visualization of the clustering result of the 117 *nr1d1*:VNP-positive cells. (f) Raw fluorescence intensity traces of the two types of *nr1d1*:VNP-positive cells in (e). (g and h) The comparison of the baseline expression (g) and relative circadian amplitude (h) between the two types of cells. The colors of the boxes correspond to (e). Two-tailed Student *t* test was applied to calculate the levels of significance between the two types of cells. ***$P < 0.001$. (i and j) Scatterplot demonstrates the relationship between baseline intensity and relative oscillation amplitude for fish1 (i) and fish2 (j), respectively. ***Pearson's correlation $P < 0.001$. The numerical values for panels c–j are in S1 Data. dpf, days postfertilization; t-SNE, *t*-distributed stochastic neighbor embedding; ZT, Zeitgeber time.

oscillations, compared to 99 out of 117 LD cells showing significant oscillations (Fig 5K). Cosine fitting with 24-hour period and shifting phases as described in our previously paper [26] was also applied to confirm this result, using cosine fitting $P < 0.05$ and relative oscillating amplitude >0.05 as the cutoff. Only 1 out of 142 DD cells showed significant oscillation, consistent with the JTKcycle result (S4A–S4C Fig). Time-lapse imaging in every hour at 5 dpf of DD cells further confirmed this result (S32 Movie, Table K in S2 Data). In fact, no oscillating cells were identified using JTKcycle adjusted $P < 0.05$ and absolute amplitude >100 as the cutoff (S4D–S4G Fig). Therefore, our result suggests that the cellular circadian oscillations in DD fish were severely dampened rather than desynchronized during the development.

## Discussion

The mouse version of *nr1d1*:VNP was originally developed by Nagoshi and colleagues in NIH 3T3 cells. VNP was chosen for its high fluorescence intensity and high folding efficiency [27]. With this reporter, they were able to show that each NIH 3T3 fibroblast cell has a self-sustained circadian oscillator that can be entrained by serum shock. Also using this reporter, two independent studies have demonstrated the tight coupling between circadian rhythm and cell cycle in proliferating NIH 3T3 cells [28,29]. But the application of *nr1d1*:VNP in circadian research has so far been limited in cell lines. In our study, we constructed live zebrafish carrying *nr1d1*:VNP fluorescence reporter, and for the first time, we can monitor the circadian gene expression at the single-cell resolution in zebrafish larvae. Using this reporter fish, we revealed the interplays among circadian clock, cell type–specific development, and LD cycle. It is noted that caveats of using a transgenic technology rather than a site-specific knock-in strategy may exist, such as the position effects of the reporter insertion. As fluorescent, ubiquitination-based cell cycle indicator (FUCCI) lines to label different stages of cell cycle are available in zebrafish [30], one can cross *nr1d1*:VNP fish with cell-cycle reporter lines to examine the relationship between circadian clock and cell cycle in vivo. By crossing fish lines containing cell type–specific fluorescent markers with our *nr1d1*:VNP fish, one can conduct cell type–specific imaging of circadian rhythm, as we have demonstrated for rod-like cells in the pineal gland. Larval zebrafish has also been used in drug screening for chemical compounds affecting circadian rhythm and sleep [31]. Our *nr1d1*:VNP fish could be used to investigate the effect of selected drug targets on the synchronization of single-cell oscillators in vivo. With advanced microfluidic systems to capture and recapture larval zebrafish, one can conduct more intense imaging on the same fish to achieve higher temporal resolution while minimizing the perturbations to the fish [32]. Using a newly developed movable platform to track zebrafish larvae and a novel volume imaging system, live imaging of cellular circadian rhythm on free-moving zebrafish will become possible [33]. In summary, our single-cell circadian reporter zebrafish line will have broad applications in circadian research.

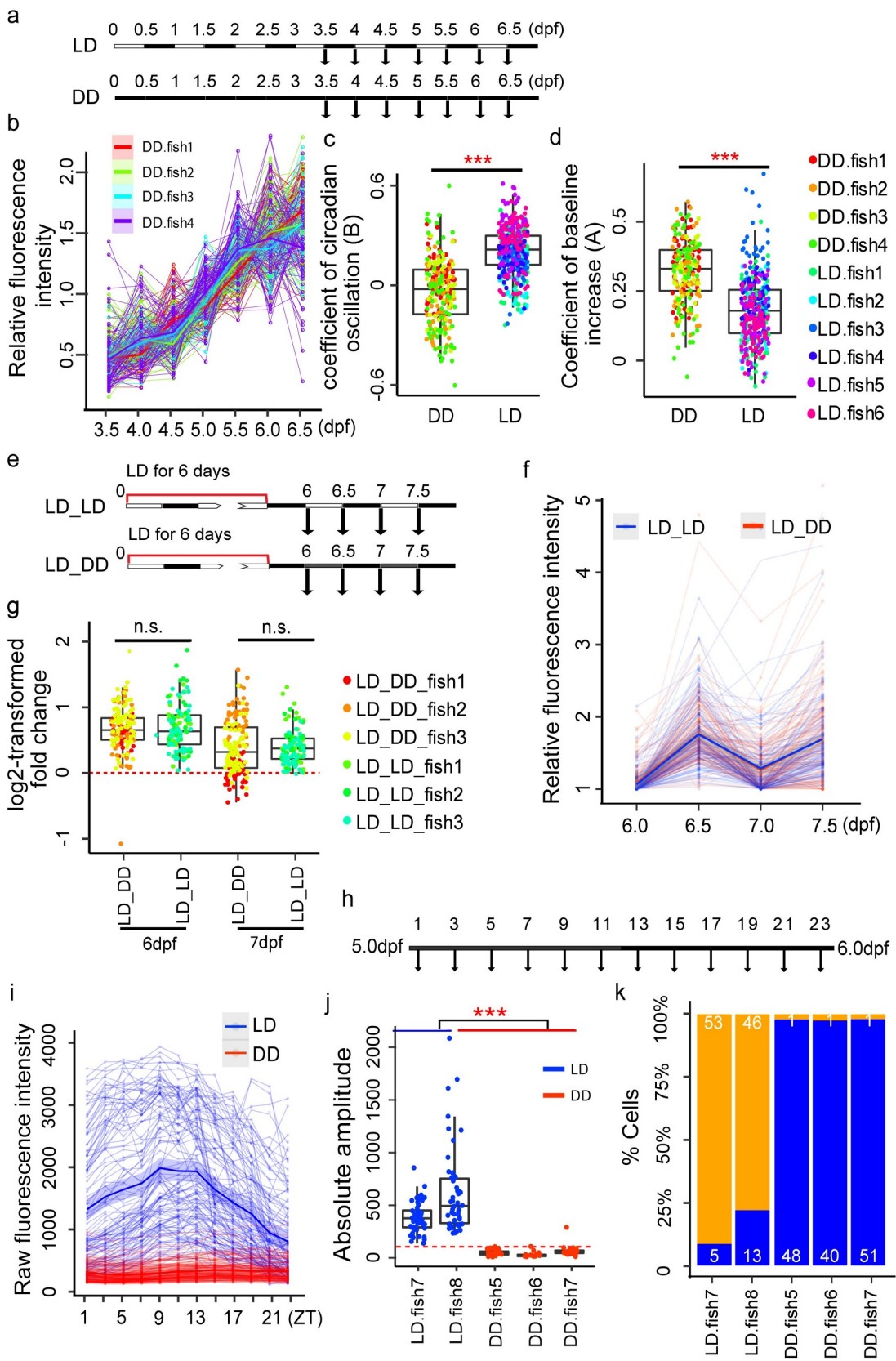

**Fig 5. LD cycle is essential for *nr1d1* oscillation.** (a) Experimental design to investigate the effect of light on the onset of circadian clock development. (b) Expression patterns of all cells under DD condition during development (four fish). Each thin line represents one cell. Each thick line represents the loess-smoothed curve of all cells in each individual fish. The shaded area shows the 95% confidence level of the smoothed curve. (c) Comparison of oscillating coefficient (B) between LD (six fish) and DD (four fish) cells. Two-tailed Student *t* test was applied to calculate the levels of significance between the two types of cells. ***$P < 0.001$. (d) Comparison of developmental coefficient (A) between LD (six fish) and DD (four fish) cells. Two-tailed Student *t* test was applied to calculate the levels of significance between the two types of cells. ***$P < 0.001$. (e) Experimental design to examine the single-cell circadian clocks under DD after transferred from LD (LD_DD). (f) Expression patterns of LD_LD (three fish) and LD_DD cells (three fish) from 6.0 to 7.5 dpf. Each thin line represents one cell and each dot represents raw fluorescence intensity. Thick lines represent the loess-smoothed curves for all the LD_DD cells in red and LD_LD cells in blue, respectively. The shaded areas show the 95% confidence level of the smooth curve. (g) Log2-transformed dusk–dawn ratios of VNP fluorescence intensities of all the cells in (f). Each data point represents one cell and the colors correspond to different fish. The red dashed line represents $y = 0$. n.s. represents no significant difference in dusk–dawn ratios between different time points (two-tailed ANOVA test). (h) Experimental design to examine the expression pattern of DD cells at higher temporal resolution. (i) Expression patterns of LD (two fish) and DD cells (three fish) across one day at 2-hour resolution. Each thin line represents one cell and each dot represents raw fluorescence intensity. Thick lines represent the loess-smoothed curves for all the LD in blue and DD cells in red, respectively. The shaded areas show the 95% confidence level of the smooth curve. (j) Comparison of the absolute amplitudes of LD and DD cells in (i) from JTKcycle. The red dashed line represents $y = 100$. (k) Percentages of oscillating cells (JTKcycle adjusted $P < 0.05$ and absolute amplitude >100) in each LD and DD fish. The orange bars represent the percentage of oscillating cells, while the blue bars represent the percentage of non-oscillating cells. The numerical values for panels b, c, d, f, g, i, j, and k were in S1 Data. DD, constant dark; dpf, days postfertilization; LD, light–dark; VNP, Venus-NLS-PEST; ZT, Zeitgeber time.

It has been shown that physiological and behavioral rhythms in zebrafish appear during the larval stages of zebrafish [34–36]. In our study, the expression of *nr1d1* appeared gradually in development and stepwise in different brain regions. As the major component of the second negative feedback loop in circadian clock, *nr1d1* is directly regulated by BMAL1 and represses the expression of *bmal1* [1]. The onset of *nr1d1* expression may represent the maturation of a functional circadian clock during development. Our single-cell study revealed the developmental trajectory of *nr1d1* expression varies greatly among individual cells, and the onset of circadian rhythm is dependent on early light exposure. Ziv and colleagues have shown that *per2* expression can be induced by light and is necessary for light-dependent onset of circadian oscillation of *aanat2* in the pineal gland [37]. In addition, *otx5*, a TF constantly expressed in circadian cycle, has been shown to regulate the expression of *nr1d1* in the pineal gland [38]. It is likely that *otx5/crx* are responsible for cell type–specific developmental components of *nr1d1* expression. Nagoshi and colleagues have suggested that the robust expression of *Nr1d1*: VNP in mouse NIH 3T3 cells is due to high transcriptional activity of *Nr1d1* promoter in these cells, as *Nr1d1* is more highly expressed in tissues such as liver compared to other core clock genes such as *Bmal1* [27]. In our case, the presence of *otx5* and *crx* binding sites in *nr1d1*:VNP promoter may have given rise to its robust expression in pineal photoreceptors and cells in other brain regions. Therefore, one can replace *otx5* and *crx* binding sites by those of other cell type–specific TFs in *nr1d1*:VNP to report the circadian rhythm in cell types of interest. This will enable single-cell imaging of circadian clock in a broad range of cell types in live zebrafish.

Several studies have shown that, in the absence of light, the overall circadian oscillation of gene expression was absent in developing zebrafish embryos [10,20,37]. But it is unclear if this is due to the lack of initiation or synchronization of cellular clocks. By imaging *per1-luc* transgenic fish, Dekens and colleagues suggested that cells in the embryos kept under constant environmental conditions may undergo asynchronous oscillations [20]. Ziv and colleagues also proposed that the expression of *aanat2* is asynchronous in individual pineal cells under constant darkness [37]. Carr and Whitmore showed that light-sensitive cells of the zebrafish DAP20 cell line kept under long-term darkness became desynchronized, and light can synchronize the clocks in these cells [11]. To our surprise, we did not observe asynchronous oscillations of *nr1d1*:VNP among individual cells in our reporter fish raised under constant

darkness. In Dekens and colleagues' study, separate zebrafish siblings fixed at circadian time (CT) 3 and CT15, respectively, under DD conditions have an intermediate number of cells expressing *per1* at the RNA level [20]. Similarly, Ziv and colleagues observed an average level of *aanat2* in the zebrafish embryos kept in constant darkness compared with the fish kept in LD cycle. However, these observations are not the direct evidence for asynchronous oscillations, as the gene expression was not traced over time in the same fish. Instead, they can be explained by the highly variable baseline gene expression between single cells within the same fish observed in our study. The difference between our study and Carr and Whitmore's study may reflect the difference between in vivo versus in vitro systems. During our time of imaging, zebrafish embryos are undergoing rapid changes of cellular states through differentiation and proliferation, which may preclude the existence of a self-sustained cellular clock. The zebrafish cell line used by Carr and Whitmore is a stable transfected cell line and can harbor free-running autonomous clocks as demonstrated in some stable mammalian cell lines [39]. However, it has also been shown that mammalian induced pluripotent stem cells (iPSCs) and ESCs are non-rhythmic [14,40], suggesting that the difference of cellular states plays an important role in the state of cellular clocks. Our study thus suggests that early light exposure in zebrafish larvae has a dramatic effect on the development of self-sustained cellular clocks, demonstrating the utility of the single-cell circadian zebrafish reporter. As *nr1d1* is mainly involved in the second circadian feedback loop, one cannot rule out the possibility that the development of cellular circadian oscillations that we observed in this study is specific to this second loop. Therefore, it will be important to further investigate the single-cell expression of other circadian clock genes such as *Per* family genes during development in future studies.

## Methods

### Ethics statement

All experiments with zebrafish were done in accordance with protocols approved by the Animal Use Committee of Institute of Neuroscience, Chinese Academy of Sciences (permission number NA-045-2019).

### Construction of the *nr1d1*:VNP (Venus-NLS-PEST) expression vector

The zebrafish *nr1d1*:VNP reporter plasmid was constructed as follows. The mouse *Nr1d1*:VNP reporter plasmid was generously provided by Emi Nagoshi [27] (Department of Genetics and Evolution, Sciences III, University of Geneva). The plasmid carries an intact VNP sequence. VNP is a nuclear fluorescent protein with a high folding efficiency and short half-life [28]. The zebrafish *nr1d1*:VNP plasmid was obtained by homologous recombination with four cassettes. The targeting cassette was generated by four-step PCR amplification. The first PCR step was performed to amplify the tol2-polyA-ampicillin gene with an additional 3′ sequence that is homologous to the 5′ end region of zebrafish *nr1d1* promoter (template: 394-tcf-lef-TuboGFPdest1, forward primer: 5′-GCTTCTGCTAGGATCAATGTGTAG-3′, reverse primer: 5′-CCTATAGTGAGTCGTATTACCAACTT-3′). The second and third PCR steps were performed to amplify the promoter of zebrafish *nr1d1* gene (6.2 kbp) (template sequences from zebrafish genome: second PCR forward primer: 5′- TACGACTCACTATAGGTTTTCCACCCGCTGGGCTGCCTCTCACGTG-3′, reverse primer: 5′-TATTCCTTGCTTCCGTCTGTAGTGCCCAAT-3′; third PCR forward primer: 5′-GGCAGACATCACGGGTTAAGACACAGTGTT-3′; reverse primer: 5′-CAGCAGGCTGAAGTTAGTAGCTCCGCTTCCTCCCGGAGGGGATGGTGGGAATGATGATCC-3′). The fourth PCR step was performed to amplify the cassette of VNP (template sequences from mouse *Nr1d1*:VNP reporter plasmid forward primer: 5′-ACTAACTTCAGCCTGCTGATGGTGAGCAAGGGCGAGGA-3′, reverse primer: 5′-CTACACAT

TGATCCTAGCAGGAGCACAG-3′). The 3′ arm of the first PCR was homologous to the 5′ arm of the second PCR, the 3′ arm of the second PCR was homologous to the 5′ arm of the third PCR, the 3′ arm of the third PCR was homologous to the 5′ arm of the fourth PCR, and the 3′ arm of the fourth PCR was homologous to the 5′ arm of the first PCR. The vector construct was verified by sequencing. Fig 1A shows the schematic of the *nr1d1*:VNP construct design. The structure of the 6.2-kbp *nr1d1* promoter containing multiple E-box, RRE, and pineal-specific elements was also described.

## Generation of transgenic Tg(*nr1d1*:VNP) zebrafish reporter line

Generation of transgenic fish using the Tol2 system was carried out according to the published protocol [41]. Constructs were injected together with capped RNA encoding transposase (10 ng/μL each of DNA and RNA) into fertilized eggs at the one-cell stage. The injected fish (F0 generation) were raised and screened for integration of the transgene into the germline. Isolation of transgene-positive progeny (F1) was carried out using EGFP imaging using fluorescence stereomicroscpe (Olymous, SZX16). For in vivo imaging, *Tg(nr1d1*:VNP*)* zebrafish was crossed with a pigmentation mutant strain (*casper* mutant), which is transparent in the whole brain except the eyes.

## Generation of transgenic Tg(lws2:nfsB-mCherry) and Tg(xops:nfsB-mCherry) zebrafish reporter line

The 1.77-kbp lws2 and 1.38-kbp xops (Xenopus rhodopsin) promoter were amplified by PCR. Specific primers for the lws2 promoter: 5′-GGCCAGATGGGCCCTGTTGTGCACCAGAT CTGAGT-3′ and 5′-TGGTCCAGCCTGCTTTTTGGAAACCCTGAAGATCA-3′. The xops promoter (1,370 bp) was amplified from pFIN-XOPS-tdTOMP (Semple-Rowland SL and colleagues, 2010, addgene #44359) using forward (5′-TATAGGGCGAATTGGGGCCGCAGAT CTTTATACATTGC-3′) and reverse (5′-CCGGTGGATCCCAAACCCTCGAGATCCCTA GAAGCCTGTGAT-3′) primers containing homogenous recombination sites, respectively. The products were subcloned into pTol-uas:nfsB-mCherry plasmid to substitute the uas promoter by using homologous recombination kit (ClonExpress MultiS One Step Cloning Kit, C113; Vazyme, China). The pTol-lws2:nfsB-mCherry plasmid and pTol-xops:nfsB-mCherry were co-injected into AB embryos with Tol2 transposase mRNA at the one-cell stage. The mCherry signal was screened to identify F1.

## Zebrafish husbandry

Adult zebrafish (NCBI taxonomy ID: 7955) were raised and maintained in fully automated zebrafish housing systems (Aquazone, Tzofit, Israel; temperature 28˚C, pH 7.0, conductivity 300 mS) under 14-hour/10-hour LD cycles, and fed with paramecium twice a day. Larvae were fed twice a day starting from 5 dpf. For the experiments under normal LD condition, embryos were produced by natural spawning in the morning and raised in egg-water containing methylene blue (0.3 ppm) in a light-controlled incubator under 12-hour/12-hour LD cycles at 28˚C. ZT0 is defined as the time when the lights are turned on (9 AM). For the experiments under DD condition, embryos were collected within 30 minutes after birth and put into a black box in a dark incubator at 28˚C. All the experimental protocols were approved by the Animal Use Committee of Institute of Neuroscience, Chinese Academy of Sciences.

We use Tg(*nr1d1*:VNP), Tg(*aanat2*:mRFP), Tg(*lws2*:nfsB-mCherry), Tg(*xops*:nfsB-mCherry), and Tg(*her4*:DsRed) zebrafish lines, which were maintained on an AB background. Sex includes male and female. We used the larva fish aged from 3.5 dpf to 7.5 dpf.

## In vivo imaging

Fish were randomly selected to be imaged under LD or DD. We were not blinded to LD and DD fish group allocation. For time-lapse imaging every two hours lasting for one day starting at ZT0 (Figs 1G, 4 and 5I–5K), 5.0 dpf zebrafish larvae were anesthetized with MS-222 (Sigma-Aldrich) (0.01%–0.02%) and mounted in low melting point agarose (1.3%–1.5%). The fish were maintained in a chamber with circulating egg-water without anesthesia at temperature of 28 ± 0.5˚C. For imaging from 3.5 dpf to 6.5 dpf every 12 hours at ZT0 and ZT12 (Figs 1E, 3 and 5A–5G) and single time point imaging (Fig 2C and 2F), the zebrafish larvae were anesthetized at each time point with MS-222 (sigma-Aldrich) (0.01%–0.02%) and embedded in low melting point agarose (0.8%–1.0%) before imaging. Three to four fish were imaged in each imaging session that lasts for one half to one hour. Fish were released immediately after the end of each imaging session and in a free-moving state between image sessions. All fish were imaged from dorsal view using two-photon microscope (Olympus) as described previously [42]. Imaging was performed using a 25× (numerical aperture [NA] = 1.05) water-immersion objective (Olympus). Excitation was provided by a Ti:Sapphire femtosecond pulsed laser system (Coherent) tuned to 900 nm, which allowed efficient simultaneous excitation of Venus fluorescent protein. Laser power was set to 33–35 mW for pineal imaging and 44.4 mW for whole-brain imaging. We used two-photon excitation microscopy for two reasons, as suggested by Carvalho and Heisenberg [43]. First, the near-infrared wavelength has undetectable effects on fish physiology and behavior. Second, as the two-photon excitation is only achieved near the focal plane, it minimizes photo bleaching and phototoxicity. The sample size estimate is based on our previous studies. Two fish under LD condition were imaged for the whole-brain imaging (Fig 1E). One fish under LD condition was imaged for one day every hour (Fig 1G). Two fish under LD condition and three fish under DD condition were imaged for one day every 2 hours (Figs 4 and 5I–5K). Six fish under LD condition and four fish under DD condition were imaged from 3.5 dpf to 6.5 dpf every 12 hours (Figs 1E, 3A–3E and 5A–5G). Three fish under LD_LD condition and three fish under LD_DD condition were imaged from 6.0 dpf to 7.5 dpf every 12 hours (Fig 5F). Six Tg(*xops*:nfsB-mCherry) fish (Fig 3F–3H) were imaged from 3.5 dpf to 6.5 dpf every 12 hours.

## scRNA-seq and data analysis

Larval heads (6-dpf) were dissected on dissection medium (DMEF/F12 with 2% 100X penicillin-streptomycin) and pineal regions were enriched by pipetting the pineal with the whole brain into a tube placed on ice. Then the dissociated cells were obtained according to the protocol from Miguel A. Lopez-Ramirez and colleagues [44]. First, add 300 uL papain solution to the dissected tissue at 37˚C in water heater for 15 minutes, with gentle pipetting during digestion. Then, stop the digestion by adding 1.2 mL of washing solution (650 μL gluose [45%], 500 μl HEPES [1 M], and 5 mL FBP were added into 93.85 mL DEBS 1X) and wash the cells twice using washing solution. In the end, sterilize the cells using a 40-um pore size filter. To count the living cells, we stained cells using trypan blue and counted the cells using a hemocytometer. The resulting single cell suspension was promptly loaded on the 10X Chromium system using Chromium Single Cell 3' Reagents v2. The barcoded cDNA library was then sequenced on the Illumina NovaSeq 6000 system.

For data analysis, raw sequencing data were converted to matrices of expression counts using the cellranger software provided by 10X Chromium; zebrafish reference transcriptome (ENSEMBL Zv11, release 95) was used as reference genome. The gene expression matrix was then loaded into Seurat package [45] in R for the following analysis. Cells with less than 500 genes or a percentage of mitochondiral genes > 0.02 were excluded from the following

analysis. The graph-based method from Seurat was used to cluster the left cells (6,514 cells). The PCA was selected as the dimensional reduction technique in the construction of the shared nearest-neighbor (SNN) graph. The first 50 PCs were then used in an SSN clustering, with the resolution equal to 1. A *t*-distributed stochastic neighbor embedding (t-SNE) clustering was performed on the scaled matrix (with highly variable genes only) to obtain a 2D representation of the cell states. Finally, 26 clusters were identified. They were manually annotated by comparing the marker genes with those from whole-brain scRNA-seq data in adult zebrafish [21].

## Image analysis

The image data were first converted to 'nrrd' format using a customer Fiji [46] macro code for downstream analysis. Time series of 3D images in 'nrrd' format for each fish were then aligned by CMTK toolkit (parameters: -awr 01 -l fa -g -T 8 -X 10 -C 1 -G 20 -R 2 -E 1 -A '—accuracy 5—dofs 12' -W '—accuracy 5' for the alignment of the pineal gland; -ar 01 -l a -A '—exploration 50—accuracy 5—dofs 12' for the alignment of whole brain) to facilitate the single cell tracing. TrackMate [23], an open source Fiji [46] plug-in, was applied to perform the single-cell nucleus identification and tracing on 3D images over time. Briefly, Laplacian of Gaussian (LoG) filter with a sigma suited to the estimated blob diameter (5.5 μm) was applied to detect the single-cell nuclei in each 3D image. Then HyperStack Displayer was applied to visualize the identified spots, which allows manual editing afterwards. A nearest-neighbor algorithm (parameter: maximal linking distance 5.5 μm) was applied to trace each single cell. After the automatic processing, the identification and tracing of each cell were manually curated using the manual tracking tools of TrackMate [23]. Combining the automated and manual tracking approaches can significantly increase the detection accuracy of single cells. In the end, the cell position and mean image intensities were exported for the following statistical analysis. For the whole-brain data, the pineal gland, the optic tectum, and the cerebellum were first manually extracted, then single cells were traced using manual tracking tools of TrackMate [23], and mean intensities of each brain region were used for the following analysis. The cell position and mean intensities for all the fish analyzed were listed in Tables B–J in S2 Data.

## Three-dimensional reconstruction of zebrafish pineal gland

To systematically analyze the 3D distribution of the *nr1d1*:VNP-positive cells, a common 3D pineal gland structure was reconstructed by averaging six zebrafish pineal glands at 5.5 dpf. First, we manually reconstructed the 3D pineal structure of six individual 5.5 dpf zebrafish pineal glands, taking advantage of the clear pineal gland boundary in two-photon imaging. Secondly, all 3D pineal gland structures from different fish were aligned using rigid transformation using CMTK (parameters: -ar 01 -l a -A '—accuracy 5—dofs 12'). Thirdly, the signals from aligned 3D pineal glands were added and smoothed by Gaussian blur (Sigma = 30) using Fiji package [46]. In the end, a common 3D pineal gland structure was obtained by implementing Li's minimum cross entropy thresholding method using Fiji package [46].

The cells of each pineal gland were then registered to the common 3D pineal gland using the same transform matrix for the alignment of the 3D pineal gland structure. The 'ks' package in R (https://www.R-project.org/) was applied to calculate the cell density distribution. The common zebrafish pineal structure and *nr1d1*:VNP cell density distribution were visualized by NeuTube software [47] (Fig 2D).

## Statistical analysis

All the statistical analysis was performed using the computing environment R (https://www.R-project.org/). The fluorescence intensities were log2-tranformed before fitting. Relative

amplitude was defined as absolute amplitudes divided by means. For the time-lapse imaging data from 5.0 dpf to 6.0 dpf every half hour, both JTKcycle (from MetaCycle package in R) and cosine fitting were applied to analysze the data. JTKcycle adjusted $P < 0.05$ and absolute amplitude $>100$ were used as the cutoff to define rhythmic cells. The circadian phase and amplitude of each cell were also calculated by fitting the mean intensity of each cell to cosine functions with a 24-hour period and shifting phases as described previously [26]. R package 'circular' was applied to calculate the average circadian peak (mean.circular) and Kuramoto's order parameter R (Rho.circular). For the imaging data of developmental time course from 3.5 dpf to 6.5 dpf, the fluorescence intensities were first normalized by dividing the mean intensity. A regression model combining a stepwise function denoting the circadian effect and a linear function denoting the developmental trend were used to fit the data. Namely, $F(x) = Ax + B^*(-1)^{(x+1)} + C$, A denotes the slope of developmental trend, B denotes the amplitude of circadian oscillation, and C denotes the baseline level of fluorescence intensity. Nonlinear least squares (nls) were applied to determine these three parameters and the level of significance of each coefficient. R package 'Seurat' was applied to perform t-SNE analysis and visualization of clusters. To classify LD cells imaged every 2 hours, all the 12 principle components were used and resolution was set to 0.4.

### Real-time qPCR

For each time point, A total of 15 to 20 fish were separated into 2 tubes equally. Here, the sample size estimate is based on our previous studies. Trizol (Invitrogen) was immediately added to each tube. Then, total RNA was isolated from zebrafish larvae by Trizol (Invitrogen) according to the manufacturer's protocol. Total RNA quantities were measured by a Nanodrop spectrometer (Nanodrop 2000). Five hundred nanograms of total RNA was reverse transcribed using HiScript II Q RT SuperMix for qPCR (+gDNA wiper) (vazyme) according to the manufacturer's protocol. Two microliters of RT product (1:10 diluted) and 10 μL of SYBR Green Master Mix (vazyme) were used in qPCR on an ABI StepOne Plus System (Thermo Fisher) according to the manufacturer's protocol. The specificity of PCR was checked by melting curve analysis. In every qPCR assay, *eef1a1* was used as the control gene for any significant bias of starting materials across samples. Primers used in this study were listed in Table L in S2 Data.

## Supporting information

**S1 Fig.** (a) Plot of real-time PCR results of *nr1d1* and *per1b* during early development (from 26 hpf to 68 hpf). Each time point is the pool of 10–20 fish. (b) Raw fluorescence values of all the traced *nr1d1*:VNP cells at 1-hour intervals for one day. (c) Circadian phase distributions of the 48 *nr1d1*:VNP-positive cells in b. (d) Fluorescence images show the co-expression of *nr1d1*:VNP (nuclear signal) and *her4*:DsRed (cytoplasmic signal) in the zebrafish pineal gland. hpf, hours post fertilization.
(EPS)

**S2 Fig. Log2-transformed dusk–dawn ratios of VNP fluorescence intensities at different time points during development (i.e., 4.5 dpf/4.0 dpf, 5.5 dpf/5.0 dpf, 6.5 dpf/6.0 dpf).** (a) Each data point represents one cell and the colors correspond to different fish. The red dashed line represents $y = 0$. n.s. represents no significant difference in dusk–dawn ratios between different time points (two-tailed ANOVA test). (b) Barplot shows the number of cells with dusk > dawn (blue color, i.e., the number of cells above the red dash line in b) or dusk < dawn (red color, i.e., the number of cells below the red dash line in b). dpf, days postfertilization; VNP, Venus-NLS-PEST.
(EPS)

**S3 Fig.** The comparison of the circadian phase between the two types of cells in Fig 4E: (a) fish1, (b) fish2. The color of the boxes corresponds to Fig 4E. Two-tailed Student *t* test was applied to calculate the levels of significance between the two types of cells. n.s. represents $P > 0.05$.
(EPS)

**S4 Fig. LD cycle is essential for *nr1d1* oscillation.** (a) Comparison of the relative amplitudes of LD and DD cells in Fig 5I from cosine fitting. The red dashed line represents $y = 0.05$. (b) Percentages of oscillating cells (cosine fitting $P < 0.05$ and relative amplitude $> 0.05$) in each LD and DD fish. The orange bars represent the percentage of oscillating cells, while the blue bars represent the percentage of non-oscillating cells. (c) Single-cell tracing results of all 142 *nr1d1*:VNP-positive cells in three zebrafish pineal glands raised under DD condition. The blue dots represent the original fluorescence signals, while the solid red line represents the smoothed curve fitted by the cosine functions. (d) Expression patterns of LD (1 fish) and DD cells (1 fish) across one day at 1-hour resolution. Each thin line represents one cell and each dot represents raw fluorescence intensity. Thick lines represent the loess-smoothed curves for all the LD in blue and DD cells in red, respectively. The shaded areas show the 95% confidence level of the smooth curve. (e) Comparison of the absolute amplitudes of LD and DD cells from JTKcycle (1-hour resolution). The red dashed line represents $y = 100$. (f) Percentages of oscillating cells (JTKcycle adjusted $P < 0.05$ and absolute amplitude $> 100$) in each LD and DD fish (1-hour resolution). The orange bars represent the percentage of oscillating cells, while the blue bars represent the percentage of non-oscillating cells. (g) Single-cell tracing results of all 24 *nr1d1*:VNP-positive DD cells imaged at 1-hour resolution. The blue dots represent the original fluorescence signals, while the solid red line represents the smoothed curve fitted by the cosine functions. DD, constant dark; LD, light–dark.
(EPS)

**S1 Data. Excel spreadsheet containing data values plotted in all main and supporting figures.**
(XLSX)

**S2 Data. Tables A–L. (A)** Promoter sequences (in Fastq format) used in this study to screen the circadian reporters in zebrafish. **(B)** Single-cell tracing result of *nr1d1*:VNP cells in the pineal gland, the optical tectum, and the cerebellum during development based on the whole-brain two-photon imaging under LD condition. The cells were tracked starting at 3.5 dpf in pineal gland, 5.5 dpf in optical tectum, and 6.5 dpf in cerebellum, respectively. Fish were imaged every 12 hours until 7.5 dpf. **(C)** Single-cell tracing result of *nr1d1*:VNP cells in the pineal gland imaged every hour under LD condition based on the two-photon imaging of the pineal gland. The fish were imaged starting at 5 dpf for one day. **(D)** Single-cell tracing result of *nr1d1*:VNP cells in the pineal gland during development under LD condition based on the two-photon imaging of the pineal gland. The fish were imaged every 12 hours from 3.5 dpf to 6.5 dpf. **(E)** Single-cell tracing result of *nr1d1*:VNP cells in the pineal gland of Tg(nr1d1:VNP) × Tg(xops:nfsB-mCherry) fish from 3.5 dpf to 6.5 dpf under LD condition. The fish were imaged every 12 hours from 3.5 dpf to 6.5 dpf. **(F)** Single-cell tracing result of *nr1d1*:VNP cells in the pineal gland imaged every 2 hours under LD condition based on the two-photon imaging of the pineal gland. The fish were imaged starting at 5 dpf for one day. **(G)** Single-cell tracing result of *nr1d1*:VNP cells in the pineal gland during development under DD condition based on the two-photon imaging of the pineal gland. The fish were imaged every 12 hours from 3.5 dpf to 6.5 dpf. **(H)** Single-cell tracing result of nr1d1:VNP cells in the pineal gland from 6.0 dpf to 7.5 dpf under LD condition based on the two-photon imaging of the pineal

gland. The fish were imaged every 12 hours from 6.0 dpf to 7.5 dpf. Those are the control cells for Table I. **(I)** Single-cell tracing result of nr1d1:VNP cells in the pineal gland from 6.0 dpf to 7.5 dpf under LD_DD condition (fish were transferred to DD at 5.5 dpf) based on the two-photon imaging of the pineal gland. The fish were raised under LD from 0 dpf to 5.5 dpf and transferred to DD at 5.5dpf. The fish were imaged every 12 hours from 6.0 dpf to 7.5 dpf. **(J)** Single-cell tracing result of *nr1d1*:VNP cells in the pineal gland imaged every 2 hours under DD condition based on the two-photon imaging in the pineal gland. The fish were imaged starting at 5 dpf for one day. **(K)** Single-cell tracing result of *nr1d1*:VNP cells in the pineal gland imaged every hour under DD condition based on the two-photon imaging in the pineal gland. The fish were imaged starting at 5 dpf for one day. **(L)** Primers sequences used in RT-PCR experiment. In Tables A–K, TrackIDs are the identifiers unique for each individual fish. PositionX/Y/Z.* and Intensity.* are the *x*/*y*/*z* position and mean fluorescence intensity of the cell in each time point. DD, constant dark; dpf, days postfertilization; LD, light–dark; RT-PCR, real-time PCR.
(XLSX)

**S1 Movie. The combined image stacks of the whole brain using two-photon imaging (whole brain fish 1).** The fish was imaged from 3.5 dpf to 7.5 dpf every 12 hours (9 stacks). The fish was raised under LD condition. From left to right, top to bottom, 3.5 dpf, 4.0 dpf, 4.5 dpf, 5.0 dpf, 5.5 dpf, 6.0 dpf, 6.5 dpf, 7.0 dpf, and 7.5 dpf. dpf, days postfertilization; LD, light–dark.
(AVI)

**S2 Movie. The combined image stacks of the whole brain using two-photon imaging (whole brain fish 2).** The fish was imaged from 3.5 dpf to 7.5 dpf every 12 hours (9 stacks). The fish was raised under LD condition. From left to right, top to bottom, 3.5 dpf, 4.0 dpf, 4.5 dpf, 5.0 dpf, 5.5 dpf, 6.0 dpf, 6.5 dpf, 7.0 dpf, and 7.5 dpf. dpf, days postfertilization; LD, light–dark.
(AVI)

**S3 Movie. The combined image stacks of the pineal gland using two-photon imaging (LD fish 8).** The fish was imaged at 5.0 dpf in every hour (24 stacks). The fish was raised under LD condition. dpf, days postfertilization; LD, light–dark.
(AVI)

**S4 Movie. Confocal 3D reconstructions of zebrafish pineal gland.** Zebrafish larvae were co-labeled with *nr1d1*:VNP (blue) and *aanat2*:mRFP (red).
(AVI)

**S5 Movie. The combined image stacks of the pineal gland using two-photon imaging (LD fish 1).** The fish was imaged from 3.5 dpf to 6.5 dpf every 12 hours (7 stacks). The fish was raised under LD condition. From left to right, 3.5 dpf, 4.0 dpf, 4.5 dpf, 5.0 dpf, 5.5 dpf, 6.0 dpf, and 6.5 dpf. dpf, days postfertilization; LD, light–dark.
(AVI)

**S6 Movie. The combined image stacks of the pineal gland using two-photon imaging (LD fish 2).** The fish was imaged from 3.5 dpf to 6.5 dpf every 12 hours (7 stacks). The fish was raised under LD condition. From left to right, 3.5 dpf, 4.0 dpf, 4.5 dpf, 5.0 dpf, 5.5 dpf, 6.0 dpf, and 6.5 dpf. dpf, days postfertilization; LD, light–dark.
(AVI)

**S7 Movie. The combined image stacks of the pineal gland using two-photon imaging (LD fish 3).** The fish was imaged from 3.5 dpf to 6.5 dpf every 12 hours (7 stacks). The fish was raised under LD condition. From left to right, 3.5 dpf, 4.0 dpf, 4.5 dpf, 5.0 dpf, 5.5 dpf, 6.0 dpf, and 6.5 dpf. dpf, days postfertilization; LD, light–dark.
(AVI)

**S8 Movie. The combined image stacks of the pineal gland using two-photon imaging (LD fish 4).** The fish was imaged from 3.5 dpf to 6.5 dpf every 12 hours (7 stacks). The fish was raised under LD condition. From left to right, 3.5 dpf, 4.0 dpf, 4.5 dpf, 5.0 dpf, 5.5 dpf, 6.0 dpf, and 6.5 dpf. dpf, days postfertilization; LD, light–dark.
(AVI)

**S9 Movie. The combined image stacks of the pineal gland using two-photon imaging (LD fish 5).** The fish was imaged from 3.5 dpf to 6.5 dpf every 12 hours (7 stacks). The fish was raised under LD condition. From left to right, 3.5 dpf, 4.0 dpf, 4.5 dpf, 5.0 dpf, 5.5 dpf, 6.0 dpf, and 6.5 dpf. dpf, days postfertilization; LD, light–dark.
(AVI)

**S10 Movie. The combined image stacks of the pineal gland using two-photon imaging (LD fish 6).** The fish was imaged from 3.5 dpf to 6.5 dpf every 12 hours (7 stacks). The fish was raised under LD condition. From left to right, 3.5 dpf, 4.0 dpf, 4.5 dpf, 5.0 dpf, 5.5 dpf, 6.0 dpf, and 6.5 dpf. dpf, days postfertilization; LD, light–dark.
(AVI)

**S11 Movie. The combined image stacks of the pineal gland using two-photon imaging (xops:nfsB-mCherry fish 1).** This is Tg(nr1d1:VNP) × Tg(xops:nfsB-mCherry) fish. The fish was imaged from 3.5 dpf to 6.5 dpf every 12 hours (7 stacks). The fish was raised under LD condition. From left to right, 3.5 dpf, 4.0 dpf, 4.5 dpf, 5.0 dpf, 5.5 dpf, 6.0 dpf, and 6.5 dpf. dpf, days postfertilization; LD, light–dark.
(AVI)

**S12 Movie. The combined image stacks of the pineal gland using two-photon imaging (xops:nfsB-mCherry fish 2).** This is Tg(nr1d1:VNP) × Tg(xops:nfsB-mCherry) fish. The fish was imaged from 3.5 dpf to 6.5 dpf every 12 hours (7 stacks). The fish was raised under LD condition. From left to right, 3.5 dpf, 4.0 dpf, 4.5 dpf, 5.0 dpf, 5.5 dpf, 6.0 dpf, and 6.5 dpf. dpf, days postfertilization; LD, light–dark.
(AVI)

**S13 Movie. The combined image stacks of the pineal gland using two-photon imaging (xops:nfsB-mCherry fish 3).** This is Tg(nr1d1:VNP) × Tg(xops:nfsB-mCherry) fish. The fish was imaged from 3.5 dpf to 6.5 dpf every 12 hours (7 stacks). The fish was raised under LD condition. From left to right, 3.5 dpf, 4.0 dpf, 4.5 dpf, 5.0 dpf, 5.5 dpf, 6.0 dpf, and 6.5 dpf. dpf, days postfertilization; LD, light–dark.
(AVI)

**S14 Movie. The combined image stacks of the pineal gland using two-photon imaging (xops:nfsB-mCherry fish 4).** This is Tg(nr1d1:VNP) × Tg(xops:nfsB-mCherry) fish. The fish was imaged from 3.5 dpf to 6.5 dpf every 12 hours (7 stacks). The fish was raised under LD condition. From left to right, 3.5 dpf, 4.0 dpf, 4.5 dpf, 5.0 dpf, 5.5 dpf, 6.0 dpf, and 6.5 dpf. dpf, days postfertilization; LD, light–dark.
(AVI)

**S15 Movie. The combined image stacks of the pineal gland using two-photon imaging (xops:nfsB-mCherry fish 5).** This is Tg(nr1d1:VNP) × Tg(xops:nfsB-mCherry) fish. The fish was imaged from 3.5 dpf to 6.5 dpf every 12 hours (7 stacks). The fish was raised under LD condition. From left to right, 3.5 dpf, 4.0 dpf, 4.5 dpf, 5.0 dpf, 5.5 dpf, 6.0 dpf, and 6.5 dpf. dpf, days postfertilization; LD, light–dark.
(AVI)

**S16 Movie. The combined image stacks of the pineal gland using two-photon imaging (xops:nfsB-mCherry fish 6).** This is Tg(nr1d1:VNP) × Tg(xops:nfsB-mCherry) fish. The fish was imaged from 3.5 dpf to 6.5 dpf every 12 hours (7 stacks). The fish was raised under LD condition. From left to right, 3.5 dpf, 4.0 dpf, 4.5 dpf, 5.0 dpf, 5.5 dpf, 6.0 dpf, and 6.5 dpf. dpf, days postfertilization; LD, light–dark.
(AVI)

**S17 Movie. The combined image stacks of the pineal gland using two-photon imaging (LD fish 7).** The fish was imaged at 5.0 dpf in every two hours (12 stacks). The fish was raised under LD condition. dpf, days postfertilization; LD, light–dark.
(AVI)

**S18 Movie. The combined image stacks of the pineal gland using two-photon imaging (LD fish 8).** The fish was imaged at 5.0 dpf in every two hours (12 stacks). The fish was raised under LD condition. dpf, days postfertilization; LD, light–dark.
(AVI)

**S19 Movie. The combined image stacks of the pineal gland using two-photon imaging (DD fish 1).** The fish was imaged from 3.5 dpf to 6.5 dpf every 12 hours (7 stacks). The fish was raised under DD condition. From left to right, 3.5 dpf, 4.0 dpf, 4.5 dpf, 5.0 dpf, 5.5 dpf, 6.0 dpf, and 6.5 dpf. DD, constant dark; dpf, days postfertilization.
(AVI)

**S20 Movie. The combined image stacks of the pineal gland using two-photon imaging (DD fish 2).** The fish was imaged from 3.5 dpf to 6.5 dpf every 12 hours (7 stacks). The fish was raised under DD condition. From left to right, 3.5 dpf, 4.0 dpf, 4.5 dpf, 5.0 dpf, 5.5 dpf, 6.0 dpf, and 6.5 dpf. DD, constant dark; dpf, days postfertilization.
(AVI)

**S21 Movie. The combined image stacks of the pineal gland using two-photon imaging (DD fish 3).** The fish was imaged from 3.5 dpf to 6.5 dpf every 12 hours (7 stacks). The fish was raised under DD condition. From left to right, 3.5 dpf, 4.0 dpf, 4.5 dpf, 5.0 dpf, 5.5 dpf, 6.0 dpf, and 6.5 dpf. DD, constant dark; dpf, days postfertilization.
(AVI)

**S22 Movie. The combined image stacks of the pineal gland using two-photon imaging (DD fish 4).** The fish was imaged from 3.5 dpf to 6.5 dpf every 12 hours (7 stacks). The fish was raised under DD condition. From left to right, 3.5 dpf, 4.0 dpf, 4.5 dpf, 5.0 dpf, 5.5 dpf, 6.0 dpf, and 6.5dpf. DD, constant dark; dpf, days postfertilization.
(AVI)

**S23 Movie. The combined image stacks of the pineal gland using two-photon imaging (LD_LD fish 1).** The fish was imaged from 6.0 dpf to 7.5 dpf every 12 hours (4 stacks). The fish was raised under LD condition. From left to right, 6.0 dpf, 6.5 dpf, 7.0 dpf, and 7.5 dpf. dpf, days postfertilization; LD, light–dark.
(AVI)

**S24 Movie. The combined image stacks of the pineal gland using two-photon imaging (LD_LD fish 2).** The fish was imaged from 6.0 dpf to 7.5 dpf every 12 hours (4 stacks). The fish was raised under LD condition. From left to right, 6.0 dpf, 6.5 dpf, 7.0 dpf, and 7.5 dpf. dpf, days postfertilization; LD, light–dark.
(AVI)

**S25 Movie. The combined image stacks of the pineal gland using two-photon imaging (LD_LD fish 3).** The fish was imaged from 6.0 dpf to 7.5 dpf every 12 hours (4 stacks). The fish was raised under LD condition. From left to right, 6.0 dpf, 6.5 dpf, 7.0 dpf, and 7.5 dpf. dpf, days postfertilization; LD, light–dark.
(AVI)

**S26 Movie. The combined image stacks of the pineal gland using two-photon imaging (LD_DD fish 1).** The fish was imaged from 6.0 dpf to 7.5 dpf every 12 hours (4 stacks). The fish was raised under LD condition from 0 dpf to 5.5 dpf and was transferred to DD condition from 5.5 dpf. From left to right, 6.0 dpf, 6.5 dpf, 7.0 dpf, and 7.5 dpf. DD, constant dark; dpf, days postfertilization; LD, light–dark.
(AVI)

**S27 Movie. The combined image stacks of the pineal gland using two-photon imaging (LD_DD fish 2).** The fish was imaged from 6.0 dpf to 7.5 dpf every 12 hours (4 stacks). The fish was raised under LD condition from 0 dpf to 5.5 dpf and was transferred to DD condition from 5.5 dpf. From left to right, 6.0 dpf, 6.5 dpf, 7.0 dpf, and 7.5 dpf. DD, constant dark; dpf, days postfertilization; LD, light–dark.
(AVI)

**S28 Movie. The combined image stacks of the pineal gland using two-photon imaging (LD_DD fish 3).** The fish was imaged from 6.0 dpf to 7.5 dpf every 12 hours (4 stacks). The fish was raised under LD condition from 0 dpf to 5.5 dpf and was transferred to DD condition from 5.5 dpf. From left to right, 6.0 dpf, 6.5 dpf, 7.0 dpf, and 7.5 dpf. DD, constant dark; dpf, days postfertilization; LD, light–dark.
(AVI)

**S29 Movie. The combined image stacks of the pineal gland using two-photon imaging (DD fish 5).** The fish was imaged at 5.0 dpf in every two hours (12 stacks). The fish was raised under DD condition. DD, constant dark; dpf, days postfertilization.
(AVI)

**S30 Movie. The combined image stacks of the pineal gland using two-photon imaging (DD fish 6).** The fish was imaged at 5.0 dpf in every two hours (12 stacks). The fish was raised under DD condition. DD, constant dark; dpf, days postfertilization.
(AVI)

**S31 Movie. The combined image stacks of the pineal gland using two-photon imaging (DD fish 7).** The fish was imaged at 5.0 dpf in every two hours (12 stacks). The fish was raised under DD condition. DD, constant dark; dpf, days postfertilization.
(AVI)

**S32 Movie. The combined image stacks of the pineal gland using two-photon imaging.** The fish was imaged at 5.0 dpf every hour (22 stacks, CT3–CT24). The fish was raised under DD condition. CT, circadian time DD, constant dark; dpf, days postfertilization.
(AVI)

## Acknowledgments

We thank Prof. Emi Nagoshi (Department of Genetics and Evolution, Sciences III, University of Geneva, Geneva, Switzerland) for the mouse *Nr1d1*:VNP reporter plasmid and Prof. Yoav Gothilf (Department of Neurobiology, The George S. Wise Faculty of Life Sciences and The Sagol School of Neuroscience, Tel-Aviv University, Tel Aviv, Israel) for *aanat2*:mRFP transgenic zebrafish.

## Author Contributions

**Conceptualization:** Haifang Wang, Jie He, Yuanhai Li, Jun Yan.

**Data curation:** Haifang Wang, Xingxing Li.

**Formal analysis:** Haifang Wang, Zeyong Yang.

**Funding acquisition:** Haifang Wang, Zeyong Yang, Jun Yan.

**Investigation:** Haifang Wang, Zeyong Yang, Xingxing Li, Dengfeng Huang, Shuguang Yu.

**Methodology:** Haifang Wang, Zeyong Yang, Xingxing Li, Shuguang Yu.

**Supervision:** Jie He, Yuanhai Li, Jun Yan.

**Validation:** Haifang Wang, Xingxing Li.

**Visualization:** Haifang Wang.

**Writing – original draft:** Haifang Wang, Jun Yan.

**Writing – review & editing:** Haifang Wang, Jie He, Yuanhai Li, Jun Yan.

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
