## [Editor Report · Decision Letter 0]

22 Jul 2019

Dear Jun, 

Thank you for submitting your manuscript entitled "Single-cell in vivo imaging reveals light-initiated circadian oscillators in zebrafish" for consideration as a Methods and Resources by PLOS Biology.

Your manuscript has now been evaluated by the PLOS Biology editorial staff as well as by an Academic Editor with relevant expertise and I am writing to let you know that we would like to send your submission out for external peer review.

**Important**: Please also see below for further information regarding completing the MDAR reporting checklist. The checklist can be accessed here: https://plos.io/MDARChecklist

Please re-submit your manuscript and the checklist, within two working days, i.e. by Jul 24 2019 11:59PM.

Kind regards,

Hashi Wijayatilake, PhD,

Managing Editor

PLOS Biology

INFORMATION REGARDING THE REPORTING CHECKLIST:

PLOS Biology is pleased to support the "minimum reporting standards in the life sciences" initiative (https://osf.io/preprints/metaarxiv/9sm4x/). This effort brings together a number of leading journals and reproducibility experts to develop minimum expectations for reporting information about Materials (including data and code), Design, Analysis and Reporting (MDAR) in published papers. We believe broad alignment on these standards will be to the benefit of authors, reviewers, journals and the wider research community and will help drive better practise in publishing reproducible research. 

We are therefore participating in a community pilot involving a small number of life science journals to test the MDAR checklist. The checklist is intended to help authors, reviewers and editors adopt and implement the minimum reporting framework. 

IMPORTANT: We have chosen your manuscript to participate in this trial. The relevant documents can be located here:

MDAR reporting checklist (to be filled in by you): https://plos.io/MDARChecklist

**We strongly encourage you to complete the MDAR reporting checklist and return it to us with your full submission, as described above. We would also be very grateful if you could complete this author survey:

https://forms.gle/seEgCrDtM6GLKFGQA

Additional background information:

Interpreting the MDAR Framework: https://plos.io/MDARFramework

Please note that your completed checklist and survey will be shared with the minimum reporting standards working group. However, the working group will not be provided with access to the manuscript or any other confidential information including author identities, manuscript titles or abstracts. Feedback from this process will be used to consider next steps, which might include revisions to the content of the checklist. Data and materials from this initial trial will be publicly shared in September 2019. Data will only be provided in aggregate form and will not be parsed by individual article or by journal, so as to respect the confidentiality of responses. 

Please treat the checklist and elaboration as confidential as public release is planned for September 2019.

We would be grateful for any feedback you may have.

---

## [Decision Letter · Decision Letter 1]

22 Aug 2019

Dear Dr Yan,

Thank you very much for submitting your manuscript "Single-cell in vivo imaging reveals light-initiated circadian oscillators in zebrafish" for consideration as a Methods and Resources article at PLOS Biology. Your manuscript has been evaluated by the PLOS Biology editors, an Academic Editor with relevant expertise, and by several independent reviewers.

As you will see from the reviews (appended below), the reviewers are generally enthusiastic about the method. However, it is also clear that some additional work is needed to fully support the main conclusions, including the title conclusion, and better address the biological question. We have discussed the reviews with the Academic Editor and would strongly encourage you to better leverage this method to its full potential (e.g. Rev. 2 and 3 have requests for better time point resolution and imaging individual identified cells). We acknowledge that this is a Methods paper, so comprehensively answering the biological question is less critical as long as the results are not over-interpreted, but please do make sure the method is fully utilized in addressing this question of circadian developmental dynamics.

Overall, in light of the reviews, we will not be able to accept the current version of the manuscript, but we would welcome resubmission of a revised version that takes into account the reviewers' comments. We cannot make any decision about publication until we have seen the revised manuscript and your response to the reviewers' comments. Your revised manuscript is also likely to be sent for further evaluation by the reviewers.

Your revisions should address the specific points made by each reviewer. Please submit a file detailing your responses to the editorial requests and a point-by-point response to all of the reviewers' comments that indicates the changes you have made to the manuscript. In addition to a clean copy of the manuscript, please upload a 'track-changes' version of your manuscript that specifies the edits made. This should be uploaded as a "Related" file type. You should also cite any additional relevant literature that has been published since the original submission and mention any additional citations in your response. 

Before you revise your manuscript, please review the following PLOS policy and formatting requirements checklist PDF: http://journals.plos.org/plosbiology/s/file?id=9411/plos-biology-formatting-checklist.pdf. It is helpful if you format your revision according to our requirements - should your paper subsequently be accepted, this will save time at the acceptance stage.

Please note that as a condition of publication PLOS' data policy (http://journals.plos.org/plosbiology/s/data-availability) requires that you make available all data used to draw the conclusions arrived at in your manuscript. If you have not already done so, you must include any data used in your manuscript either in appropriate repositories, within the body of the manuscript, or as supporting information (N.B. this includes any numerical values that were used to generate graphs, histograms etc.). For an example see here: http://www.plosbiology.org/article/info%3Adoi%2F10.1371%2Fjournal.pbio.1001908#s5.

For manuscripts submitted on or after 1st July 2019, we require the original, uncropped and minimally adjusted images supporting all blot and gel results reported in an article's figures or Supporting Information files. We will require these files before a manuscript can be accepted so please prepare them now, if you have not already uploaded them. Please carefully read our guidelines for how to prepare and upload this data: https://journals.plos.org/plosbiology/s/figures#loc-blot-and-gel-reporting-requirements.

Upon resubmission, the editors will assess your revision and if the editors and Academic Editor feel that the revised manuscript remains appropriate for the journal, we will send the manuscript for re-review. We aim to consult the same Academic Editor and reviewers for revised manuscripts but may consult others if needed.

We expect to receive your revised manuscript within two months. Please email us (plosbiology@plos.org) to discuss this if you have any questions or concerns, or would like to request an extension. At this stage, your manuscript remains formally under active consideration at our journal; please notify us by email if you do not wish to submit a revision and instead wish to pursue publication elsewhere, so that we may end consideration of the manuscript at PLOS Biology.

When you are ready to submit a revised version of your manuscript, please go to https://www.editorialmanager.com/pbiology/ and log in as an Author. Click the link labelled 'Submissions Needing Revision' where you will find your submission record. 

Sincerely,

Hashi Wijayatilake, PhD, 

Managing Editor

PLOS Biology

REVIEWS:

Reviewer #1: 

The study is novel in terms of approach and in term of findings. A new tool has been developed, using an established clock reporter, to monitor circadian clock function at a single-cell level in an intact animal (the zebrafish larvae) and to study the ontogeny of the circadian clock. The new finding is that the cellular clocks require light exposure in order to start ticking and this is different than what was previously thought. In addition to these main finding a description of the sequential development of the clock in the brain (starting in the pineal gland) and the two different types of photoreceptor cells with differential clock functions are new and important. 

In spite of a very complex subject and complex methodology the manuscript is clear and easy to follow. 

Major comments

1. The main finding that light is required for the initiation of cellular clocks is different from what has been previously described and suggested. The groups of Whitmore and Foulkes have shown, in zebrafish cell lines, that a pulse of light synchronizes the period of unsynchronized cellular oscillators. The group of Gothilf, suggested that the same thing may occur in the pineal gland, but have not directly shown this. In fact in the reference used here (ref. 30) is it stated that "It is unknown whether onset reflects the initiation of the oscillator or the synchronization of many, already free running, cellular oscillators. The data presented in this study cannot entirely support or contradict either of these two alternatives." It seems that the current study solves this question for the pineal cells. Therefore, I would like to speculate that the differences between the current study and the studies in cell lines reflect cell-type differences. There could be other reasons for this contradiction. In any case, I think it is very important that the authors discuss the reasons for the differences between their findings and earlier studies.

2. The authors show that pineal cells with high basal levels of expression have low amplitude rhythms. Could this reflect a technical problem? If fluorescence has reached saturation, it would be impossible to detect rhythms in cells that express high fluorescence levels (rod-like cells in this case). I may be wrong but such saturation appears to occur in some individual traces in figure 4c. Please make sure that these individual cells that reach saturation (if indeed exist) do not affect your results and conclusion regarding the two-cell theory. 

3. nr1d1 is a clock gene and also a light induced gene (Ben Moshe et al., 2014). Is it reflected in the results? Is it being considered? 

4. I may have missed this. In the in vivo imaging that was done every 12 hours, 

larvae were anesthetized and embedded at each time point. How was it possible to detect the same cell each time?

Minor comments:

Given the main point of the paper is the cellular clocks, consider adding 'cellular' to the title: "Single-cell in vivo imaging reveals light-initiated cellular circadian oscillators in zebrafish"

Results, ' Characterization of nr1d1:VNP expressing cells'. It is important to emphasize the high levels of nr1d1 mRNA in the pineal cells (fig 2b). It strgenthens the finding using the transgene (fig 1c and fig 1e). 

Results, second paragraph. It is stated that "both genes showed higher expression level at the dawn than dusk over days (Fig. 1b)" but in figure 1b it seems that the samples were taken in mid-day and mid night.

The use of 'rod' and 'cone' cells is confusing because rods and cones are retinal cells. I suggest naming them 'rod-like' and 'cone-like'. Also, the term 'rod fish', although clear to me, is not appropriate.

What is the rod marker xops? I cannot find the primers used for xops (materials and methods) in the zf genome nor in frog genomes 

Discussion 2nd paragraph, "It was believed that physiological and behavioral rhythms in zebrafish appear during the early stages of zebrafish development [8]." This is a vague sentence. It is not clear what is 'early stages', I guess it refers to larval stages because behavioral rhythms are not seen in early stages of embryogenesis. If indeed the meaning is larval stages than the frase 'it is belived' is not accurate – it was shown. I suggest to change this sentence to " It has been shown that physiological and behavioral rhythms in zebrafish appear during the larval stages of zebrafish" Also, maybe a better reference than no. 8.

Results, first paragraph. Please indicate how were the plasmids of bmal1a/bmal2/per2/nr1d1:TGFPD1 screened. Was that performed by stable or transient expression? If stable, in which generation?

Results, second paragraph. The presence of CRX/OTX binding sites is probably the reason for the enhanced pineal expression. Please mention this.

Results, second paragraph, " scRNA-seq data also suggested that nr1d1:VNP is expressed in proliferative cells". Please emphasize that this is in the pineal gland, and at what age? This is important because it means that photoreceptors are still proliferating. 

Minor editorial points

Page 6, 2nd paragraph, epidemis should be epidermis.

Page 7. The word 'the' is missing: "…we found that VNP positive cells within the pineal gland display"

Page 8, bottom "We also observed that the cluster with higher baseline level (cluster 1) also showed". I think there is a confusion and this should be cluster 2. Please check.

Page 9, "signals in zebrafish larvae raised under the constant…" remove 'the'.

Figure 2c and 2d should be separated. Its confusing

Methods, 'in vivo imaging', bottom. Starting with "Two fish under light-dark (LD)….". It will be informative to add figure numbers.

Methods, Single cell RNA-seq and data analysis. This is written like someone's protocol.

Figure 1c. need orientation

Figure 1 legends, "The putative RRE (Nr1d1/2 binding site), E-box (Bmal1/Clock

binding site), Crx, Otx5 and Crx/Otx5 binding sites were indicated in blue rectangle,

red oval, green oval, yellow rectangle, and dark green oval respectively". There is a mistake in the order. And there are two oval greens.

Figure 2 legends, (a) should be "t-SNE visualization of brain cell clusters…"

Figure 2 legends, (b). please add: "note the highest levels in the pineal gland"

--

Reviewer #2: 

The paper by Wang et al., addresses the question of whether cellular clocks exist and function spontaneously from the initial cell divisions of embryogenesis or alternatively whether they are initialised/activated by an environmental signal (light) during development. To address this question Wang et al., have developed a reporter line nr1d1:VNP in transgenic zebrafish to report single cell dynamics of a clock driven reporter. The authors use this clock reporter line to measure single cell dynamics in all cells of the embryonic zebrafish brain/pineal from 3.5 to 6.5 dpf. They try to separate circadian and developmental dynamics. 

The authors are correct in that zebrafish are the perfect preparation to address this question. In addition, this issue has been debated greatly in the literature, with some groups promoting the spontaneous and synchronous start of the clock to those that suggest a spontaneous beginning to oscillations, but with a need for an entrainment stimulus (light) to synchronize cells. The authors are correct in that the only way to approach this problem is by single cell imaging of clock function in the living embryo, and they have generated a superb tool to test this hypothesis. 

My main problem is that I don’t feel they have succeeded in answering this question yet by the nature in which they have performed their imaging experiments. 

The authors performed imaging experiments at set intervals rather than performing continuous (or relatively continuous) imaging on identified cells. Initially only looking at two time points per day, but then every two hours over one cycle in constant darkness. Two time points is not sufficient resolution to look for possible phase differences between oscillators. And the way that the imaging is performed is akin or similar to typical molecular sampling or performing in situ hybridization experiments. 

There are two possible options in the DD experiment. a) either the clock is not oscillating in each cell, which will produce a “flat-line” in expression as seen or b) the cells are oscillating in DD but with completely different and dispersed phases, which by sampling in this way would also produce a “flat” data set, but with significant error bars, reflecting the diverse phases of the cells. Both biological states give the same apparent outcome.

In these imaging experiments, it is essential that identified cells are followed dynamically, such that one can follow the changes in expression within an individual “known” cell. In this way, either the expression levels will be flat in a given cell or may oscillate, but with random phases in the population. One cannot randomly pool the data from the population imaging. 

A less important, but significant point. When does the rhythm in nr1d1 begin in development, at a global level on an LD cycle? Multiple papers have shown that the period gene rhythm is apparent globally somewhere on the second day of development – 24 hour or so onwards. The imaging data in this study is from day 3.5 onwards, when it is known that the clock is running. If the nr1di rhythm starts latter in development, i.e. the second clock loop develops more slowly than the primary loop, then sadly this reporter is not optimal for studying when or how the clock begins. A simple in situ or qPCR data set would be very useful to address this. 

There are number of other minor comments to make. However, until the imaging is repeated with identified cells, these are of little significance. The authors have made a very powerful circadian tool and I am sure they will use it to generate a great deal of new, insightful data. However, at this point I do not think that they are quite there yet. Though many of their results are interesting. 

Regarding the tissue specific nature of their reporter. This makes this a powerful tool to study the pineal gland, and the establishment of rhythms later in other brain regions, but perhaps by using a more ubiquitous promoter they might be able to follow oscillations is earlier stages of development.

--

Reviewer #3: 

In this study, Wang et al., examined whether circadian oscillation of the pineal gland in zebrafish is stopped or desynchronized during development without LD cycles. They generated a transgenic zebrafish line carrying a destabilized fluorescent protein to monitor one of clock genes, nr1d1, at the single cell level using a two photon microscope. Since single cell time-lapse imaging of clock gene expression in vivo have never been performed, it still unclear how clock gene expression occurs under DD in zebrafish. By using this method, they confirmed Nr1d1 expression rhythms in the pineal gland and other areas under LD cycle. They found two different populations in the pineal gland. One showed rapid increase in the baseline level while the other showed robust circadian oscillations, which were corresponding to rod and non-rod cells, respectively. Importantly they found that nr1d1 expression in the pineal gland did not show circadian oscillations under constant darkness. From these results, they concluded that light exposure in early clock development initialized cellular clocks. The hypothesis is interesting and optical imaging of single cells in vivo is fascinating. However, they have not shown enough data and analysis to support their conclusion. There are some concerns for the authors to address.

Major concerns

1. The authors concluded that cellular circadian oscillations were not initiated without environmental stimuli (also the title of the manuscript). This is too strong a statement from the results. Previous studies suggested that circadian rhythms (per expression) were desynchronized in the embryos under DD. The discrepancy between these reports and the present study might be due to different clock genes. There are multiple circadian feedback loops, such as Per/Cry loop and Bmal1/Clock/Reverb-a loop. The author measured only nr1d1 expression, but if they measure per expression in single cells, circadian rhythms might be detected under DD. Without this data, it is difficult to conclude that “circadian oscillations” were not initiated under DD. In addition, the authors need to deeply discuss discrepancy between previous studies (Denkens et al. 2008; Carr et al. 2005) and the present study.

2. The authors measured nr1d1 expression every two hours from 5.0 to 6.0 dpf under DD and concluded that circadian rhythms were not detected at the single cell level (Figure 5i and k). Please clearly describe how the significance of circadian rhythms was analyzed. Cosiner method was used for rhythm analysis or other methods? This is a very important point in this paper, because low amplitude oscillation and no rhythmicity are totally different. The authors need to evaluate circadian rhythmicity in single cells by using several methods. In addition, low resolution time point data might result in false negative. The authors also need to discuss this point. 

Minor concerns

1. The authors used cosine function with 24 hours period to determine circadian phase and amplitude (p18). Since circadian period and trend is different cell by cell, they should add two more parameters “period” and “trend” to precisely determine phase and amplitude.

2. In this research, analysis of circadian rhythms in single cell level is critical. Please show nr1d1 expression data of individual cells under DD with cosine fitting data in Figure 5i similar to Figure 4c. Also it is better to show raw fluorescence data in Figure 5i to be able to directly compare LD and DD conditions.

3. The author found two clusters in the pineal gland (Figure 4e-h). These two clusters were observed under constant darkness? It seems not to be observed two clusters under DD (Figure 5i). If not, these clusters were driven by light dark cycles? Furthermore, please discuss physiological roles of these two populations in the pineal gland.

4. P5, 2nd para. The authors mentioned “both genes showed higher expression level at the dawn than dusk over days (Fig. 1b)”. But they measured these gene expression rhythms during middle of day and night, so dawn and dusk is not appropriate here.

5. P7, 1st para. The authors compared nr1d1:VNP and aanat2:mRFP, xops:nfsB-mCherry, lws2:nfsB-mCherry, or her4:Dsred reporter. Some VNP positive neurons seem not to express mRFP in Figure 2c. Please indicate the number of cells, percentage of overlap, and the number of fish. Especially, xops:nfsB-mCherry data is important because they separated rod and non-rod cells using the reporter in Figure 3.

6. Figure 3e and d; p7-8. The authors used fitting analysis for the data obtained from 3.5 to 6.5 dpf. They mentioned that “rod cells have a faster rate of developmental increase in nr1d1:VNP signals (A) but lower amplitude of circadian oscillation (B) than non-rod cells (Fig. 3f-h)”. If the data is lineally increased, (A) can be indicated as a developmental increase. But in the case of rod cells, the data seem not linear, so (A) can not simply reflect a developmental increase. They also described (B) as amplitude of circadian rhythms. However, the data were obtained from two points per day. If the circadian phase is shifted, it is plausible that (B) became small value. 

7. P9, 2nd para. It is difficult to say “the circadian expression had been absent in DD cells” from two points per day data.

8. P11, 2nd para. Dekens et al. did not use per1-luc imaging.

9. P15, 1st para. Laser power should be described like “W/mm2” not “%”.

10. P15, 1st para. Please indicate number of fish they used in figure legends to easily understand.

11. P15, 1st para. “Three fish under LD_LD condition and Three fish under LD_DD condition were….”. Second “three” should not be capital. 

12. Figure 2b and e. What is the y axis? 

13. Figure 3b. There is no information about green circles in the pictures.

14. P18. Please describe t-SNE analysis in detail.

--

Reviewer #4: 

Summary: The manuscript authored by Wang et al described beautiful recordings on how the circadian clock kicks off during the development and how it maintains after the maturation in the pineal gland, a master organ of the zebra fish clock. Backed with a state-of-the-art technology of single cell-sequencing, the authors believed that this reporter fish line can be used in a variety of stimulation conditions where the cellular clock can be real-timely monitored in vivo. This reviewer support its publication after the revision.

Major concerns:

1. The reporter line was generated by a transgenic technology rather than a site-specific knockin strategy. Even though the authors spent quite a bit space of the ms to explain/ratify the faithfulness of the reporter with the endogenous gene expression, the conclusion of each single experiment ought to be drawn in cautious. For example, the data of tracing the dynamics of nr1d1 (i.e. the clock) in each particular cell are more convincing than the comparison of among them. You can never rule out the position effects of the reporter insertion. After all, the authors, and probably people in the fish clock field, do not know how the ZfP2 regulates nr1d1 expression while nevertheless it still sits in the transgene to keep the function of ZfP1 intact (Fig1a). The populational comparison of reporter signal vs gene expression cannot be extended to single cell level. Of course this reviewer doesn't mean to assert that the reporter is useless or wrong, but it may just like the Per1-luc transgenic vs PER2::LUC knockin reporters in mammalian clock field, the limitation of this reporter should be addressed in the discussion.

2. A second major concern is that the Fig4 and 5 seem to be separated, and neither were fully investigated. This reviewer possesses a feeling that the authors has raised some interesting questions based on their observations, but stop short of answering them. However, if this is the "Methods and Resource" section of the journal designed for, I'm OK with that.

3. The authors summarized in the abstract that "light exposure in early clock development initializes cellular clocks rather than synchronizes existing individual oscillators". While in the main text and figure legend, they claimed that "Light-Dark cycle is essential to synchronous onset of nr1d1 oscillation". Please reconcile them.

Minor concerns:

There are a few typos/mispresentations in the ms. For instances: Fig1b, are the blue/red lines in right phase with x-axis? Figure legends Fig3, there are two (g) (page27).

---

## [Decision Letter · Decision Letter 2]

20 Dec 2019

Dear Dr Yan,

Thank you for submitting your revised Methods and Resources article entitled "Single-cell in vivo imaging of cellular circadian oscillators in zebrafish" for publication in PLOS Biology. I have now obtained advice from the original reviewers and have discussed their comments with the Academic Editor. As you can see, three of the reviewers are satisfied with the revision. Reviewer 3 has some remaining requests for adding more data - I've discussed these with the Academic Editor and we encourage you to add single cell rhythms for per in DD and the other data requested if you have this data available. Otherwise please textually discuss the issues raised by Reviewer 3 in the manuscript. Overall, based on the reviews, we will probably accept this manuscript for publication, assuming that you will modify the manuscript to address the remaining points raised by the reviewers. Please also make sure to address the data and other policy-related requests noted at the end of this email.

We expect to receive your revised manuscript within two weeks. Your revisions should address the specific points made by each reviewer. In addition to the remaining revisions and before we will be able to formally accept your manuscript and consider it "in press", we also need to ensure that your article conforms to our guidelines. A member of our team will be in touch shortly with a set of requests. As we can't proceed until these requirements are met, your swift response will help prevent delays to publication.

*Copyediting*

*Published Peer Review History*

*Early Version*

*Submitting Your Revision*

Sincerely,

Hashi Wijayatilake, PhD, 

Managing Editor

PLOS Biology

ETHICS STATEMENT:

The Ethics Statements in the submission form and Methods section of your manuscript should match verbatim. Please ensure that any changes are made to both versions.

-- Please include the full name of the IACUC/ethics committee that reviewed and approved the animal care and use protocol/permit/project license. **Please also include an approval number.**

-- Please include the specific national or international regulations/guidelines to which your animal care and use protocol adhered. Please note that institutional or accreditation organization guidelines (such as AAALAC) do not meet this requirement.

-- Please include information about the form of consent (written/oral) given for research involving human participants. All research involving human participants must have been approved by the authors' Institutional Review Board (IRB) or an equivalent committee, and all clinical investigation must have been conducted according to the principles expressed in the Declaration of Helsinki.

DATA POLICY:

Figs. 1BEG, 2BE, 3C-G, 4C-J, 5BCDFGIJK, S1ABC, S2AB, S3AB, S4ABC

- Please ensure that your Data Statement in the submission system accurately describes where your data can be found. Your Data Statement currently says: 

"All the imaging data were included in the supplementary data. And the single cell RNA-seq data has been deposited on GEO (GSE134288)." The GEO dataset is currently private and is scheduled to be released on Jul 01, 2020. Please release this now.

REVIEWS:

Reviewer #1: 

Thank you for addressing all my concerns adequately

--

Reviewer #2: 

The revised manuscript PB-D-19-02041R2, by Wang et al, represents a significant improvement from the initial submission. Aspects of the imaging approach have been clarified and the added experiments (qPCR especially) address many of the key biological questions. I believe that this manuscript is now acceptable and worthy of publication in PLOS Biology. The study represents a significant technical achievement and generates biological data that is both interesting and intellectually stimulating. These data should be "seen" by the wider scientific community. 

The more one thinks about the data presented the more interesting it becomes and consequently the more questions it generates. But these are for future studies. One comment the authors may wish to add to their discussion is that all clock studies in zebrafish have so far focused on one of the two (or possibly three) feedback loops that make up the circadian clock system. The data in this study is believable, but it is possible that the rapid dampening in DD is something specific to the "second" loop. And the classic clock/bmal - cry/per loop might behave differently. This would explain all of the differences seen between the various zebrafish groups studying this phenomenon. It is an interesting possibility that light input, as well as developmental timing, might vary across the feedback loops…..or the clock itself might develop differentially. The future will tell. 

Congratulations to the authors on an excellent study.

--

Reviewer #3: 

In the revised manuscript, Wang et al. provided supplemental analysis and data to address the previous comments. I remain supportive of this work, but there also remain a few issues which I respectfully disagree. 

As I mentioned in the previous comments, low time resolution might result in false negative. The authors performed imaging with high time resolution (every one hour) at 5.0 to 6.0 dpf under LD (Fig 1g). But important point is DD condition. If they include LD data, I recommend them to show one hour data at DD condition to compare LD and DD condition. 

The authors provided expression pattern of each DD cells in Fig S4c as I suggested. However many cells seem to be rhythmic. It is confusing, because they concluded that single-cell oscillations were dramatically dampened in animals raised under constant darkness. Please show the single cell data obtained from DD.

Fig 5j,k and Fig S4a,b are same data. Since the author performed JTK cycle in Fig.5j,k and cos fitting in Fig S4a,b, the results must not be the same. Please check the original results.

P13 The author described as “However, it has also been shown that mammalian iPS cells are non-rhythmic…”. I recommend them to add ES cells which is also the non-rhythmic case.

As I mentioned in the previous comments, Per oscillation might be detected in single cell level under DD, even though nr1d1 expression rhythm was dampened. The authors showed nr1d1 and per1b expression rhythms under LD using qPCR data, but it could not answer this question. In addition, circadian amplitude of per1b rhythms was high as compare to that of nrldl. Without a recording of single cell level of Per expression under DD, it is difficult to answer. However it might be beyond the scope of this study. Instead of doing this experiment, please discuss this issue in the manuscript. 

As I mentioned previous comments, I suggested putting two parameters “period” and “trend” into cos fitting in two hours data (one hour data also). Because fluorescence is increased during development and period would be different cell by cell, if they use add two parameters, quality of the fitting results could be increased. I assume that some non-rhythmic neurons might be rhythmic.

--

Reviewer #4: 

The revised version has addressed all my concerns. Therefore, I support its publication asap.

---

## [Editor Report · Decision Letter 3]

10 Feb 2020

Dear Dr. Yan,

On behalf of my colleagues and the Academic Editor, Dr. Achim Kramer, I am pleased to inform you that we will be delighted to publish your Methods and Resources in PLOS Biology. 

PRESS 

Kind regards,

Krystal Farmer

PLOS Biology

on behalf of

Hashi Wijayatilake,

Managing Editor

PLOS Biology